# Negative-Guided Subject Fidelity Optimization for Zero-Shot Subject-Driven Generation

## Abstract

We present Subject Fidelity Optimization (SFO), a novel comparative learning framework for zero-shot subject-driven generation that enhances subject fidelity. Existing supervised fine-tuning methods, which rely only on positive targets and use the diffusion loss as in the pre-training stage, often fail to capture fine-grained subject details. To address this, SFO introduces additional synthetic negative targets and explicitly guides the model to favor positives over negatives through pairwise comparison. For negative targets, we propose Condition-Degradation Negative Sampling (CDNS), which automatically produces synthetic negatives tailored for subject-driven generation by introducing controlled degradations that emphasize subject fidelity and text alignment without expensive human annotations. Moreover, we reweight the diffusion timesteps to focus fine-tuning on intermediate steps where subject details emerge. Extensive experiments demonstrate that SFO with CDNS significantly outperforms recent strong baselines in terms of both subject fidelity and text alignment on a subject-driven generation benchmark.

## 1 Introduction

Recent advances in diffusion models have led to remarkable improvements in text-to-image (TTI) generation (Rombach et al., 2022; Saharia et al., 2022b; Esser et al., 2024; Labs, 2024), producing highly photorealistic images. These models have powered image editing (Hui et al., 2024; Huberman-Spiegelglas et al., 2024; Sheynin et al., 2024; Mokady et al., 2022; Hertz et al., 2023), image-to-image translation (Meng et al., 2022; Saharia et al., 2022a; Zhang et al., 2023), and, of particular interest here, subject-driven text-to-image generation (Ruiz et al., 2023; Kumari et al., 2023; Gal et al., 2023; Hu et al., 2022; Frenkel et al., 2024; Ye et al., 2023; Li et al., 2024; Patel et al., 2024; Chen et al., 2024a; Ma et al., 2024), which generates images that include the subject from a reference image while reflecting a new context described by a given text prompt. Early approaches (Ruiz et al., 2023; Kumari et al., 2023; Gal et al., 2023) require hundreds of fine-tuning iterations per subject with multiple images of the target subject, limiting scalability.

To address scalability, recent zero-shot approaches (Ye et al., 2023; Pan et al., 2024; Li et al., 2024; Wang et al., 2025; Tan et al., 2024; Zhang et al., 2025) fine-tune the pre-trained TTI model on a large triplet dataset composed of reference images, target texts, and target images in a supervised manner with the diffusion loss used in the pre-training stage. This supervised fine-tuning implicitly guides the model to mimic the target data by leveraging the reference subject image. Despite improved efficiency, these zero-shot approaches often fail to fully capture fine-grained subject details.

This limitation arises because supervised fine-tuning, which adapts a pre-trained TTI model–originally trained on a broad distribution–to a specific subject mode, relies only on target images and the mode-covering diffusion loss. The mode-covering nature in the fine-tuning stage often fails to sufficiently narrow the sampling distribution toward the mode that captures the fine details of the target subject. This is evident in Fig. 1, in which the generated subjects from the supervised fine-tuned model are visually similar but fail to capture fine-grained details. Therefore, we argue that it is necessary to explicitly incorporate *negative signals or regularization that suppress alternative modes* during fine-tuning to better focus on the desired subject attributes.

To address this, we incorporate synthetic negative targets of lower subject fidelity than positive counterparts into the fine-tuning of zero-shot subject-driven TTI models. By leveraging the subject

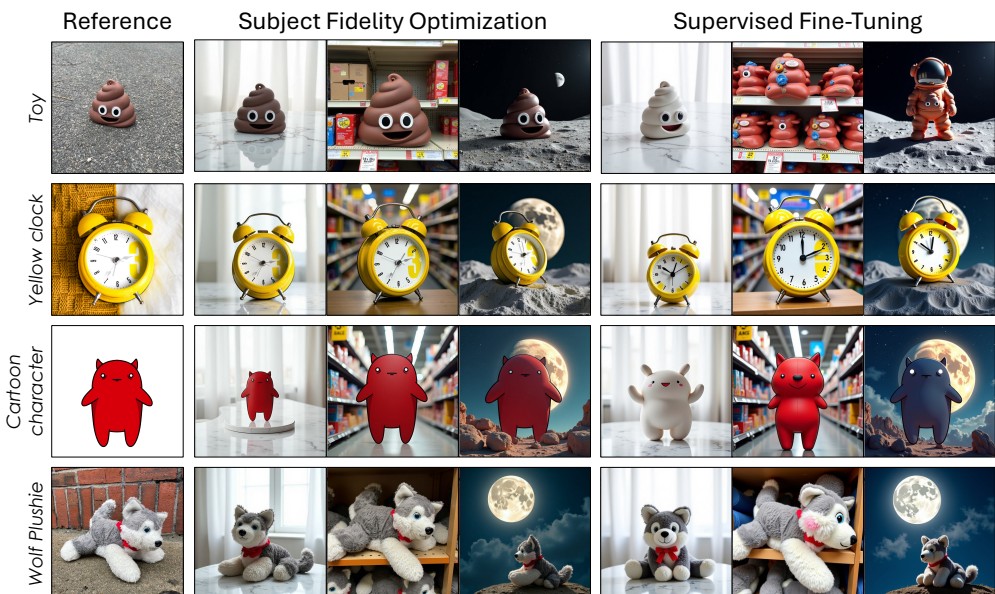

Figure 1: Our Subject Fidelity Optimization (SFO) framework improves subject fidelity in zero-shot subject-driven text-to-image generation by introducing negative targets and explicitly guiding the model regarding which aspects are desirable and which are not. The supervised fine-tuning results are obtained using our base model, OminiControl (Tan et al., 2024), and the results shown above on both sides are generated with the same seed and prompt (prompts are included in the Appendix).

fidelity gap between these targets, we propose Subject Fidelity Optimization (SFO)—a comparison-based fine-tuning strategy that explicitly guides the model to favor positive targets over negative ones. To ensure effective comparisons during SFO, we introduce Condition-Degradation Negative Sampling (CDNS), which synthesizes informative and distinguishable negative targets by intentionally degrading both visual and textual conditions. This yields negative targets with reliably lower subject fidelity and various levels of textual alignment, enabling practical and effective comparison-based learning. We further focus fine-tuning on intermediate diffusion timesteps, where subject-specific details emerge and have been identified as critical for fine-grained fidelity (Karras et al., 2022; Choi et al., 2022; Esser et al., 2024). In terms of DINO (Caron et al., 2021) and CLIP (Radford et al., 2021) scores, our SFO outperforms all baselines in subject fidelity while maintaining appropriate levels of text alignment, and it further demonstrates clear superiority in human evaluations, surpassing existing approaches. Moreover, comprehensive ablation studies validate its effectiveness and reveal the synergistic contributions of its core components.

Our contributions are summarized as follows:

- We highlight the necessity of negative targets in fine-tuning zero-shot subject-driven TTI frameworks and propose Subject Fidelity Optimization (SFO), a timestep-reweighted comparison-based fine-tuning framework.

- We introduce Condition-Degradation Negative Sampling (CDNS) for synthesizing negative targets tailored for subject-driven generation in a practical and effective manner.

- We provide empirical validation through extensive experiments and ablations, demonstrating significant improvements in subject fidelity and text alignment over existing baselines.

## 2 RELATED WORK

### 2.1 SUBJECT-DRIVEN TEXT-TO-IMAGE GENERATION

Recent progress in text-to-image (TTI) diffusion models (Rombach et al., 2022; Saharia et al., 2022b; Esser et al., 2024; Labs, 2024) has enabled high-quality image synthesis from textual

prompts. However, these models typically struggle with accurately depicting specific subjects unseen during pre-training. To tackle this, initial studies (Ruiz et al., 2023; Kumari et al., 2023; Gal et al., 2023; Hu et al., 2022; Frenkel et al., 2024) fine-tune diffusion models using a few (3–5) reference images. Despite achieving high subject fidelity, these per-subject fine-tuning methods are computationally expensive.

To overcome these efficiency limitations, zero-shot approaches bypass per-subject tuning by fine-tuning the TTI model using the same diffusion loss as in the pre-training stage while conditioning on vision-encoder embeddings (Radford et al., 2021; Caron et al., 2021; Li et al., 2023) as additional information (Ye et al., 2023; Li et al., 2024; Patel et al., 2024; Chen et al., 2024a; Ma et al., 2024; Wang et al., 2025). Yet, these embeddings primarily capture only coarse semantics, causing a loss of fine subject details in the generated image. Alternatively, JeDi (Zeng et al., 2024) and subsequent works (Shin et al., 2025; Huang et al., 2024) reframe subject-driven TTI as an inpainting task within image-set generation, achieving improved results due to the targeted fine-tuning or inherent capability of high-performing recent TTI models.

With Diffusion Transformers (DiT) (Peebles & Xie, 2023) and strong open-source TTI models (Esser et al., 2024; Labs, 2024), recent methods (Tan et al., 2024; Zhang et al., 2025) directly incorporate the latent tokens of a reference image into the input sequence of DiT and conduct joint attention over both the reference latent tokens and the text tokens. These methods require only minimal architectural adjustments and fine-tune the model in a supervised setting on a large triplet dataset using the same diffusion loss as pre-training. However, their reliance on standard diffusion losses occasionally results in incomplete preservation of subtle subject traits. This limitation motivates the investigation of alternative learning signals to further improve subject-driven TTI.

## 2.2 COMPARATIVE LEARNING SIGNALS

A number of studies explore comparison-based learning approaches that leverage both positive and negative data, extending beyond traditional supervised approaches. In contrastive learning (He et al., 2020; Chen et al., 2020), models treat an augmented view of the same image as the positive target and other images as negatives, and train to attract positive pairs closer while pushing negatives apart. Recently, preference optimization (Ouyang et al., 2022; Rafailov et al., 2023; Wallace et al., 2024) has focused on training generative models to favor human-preferred data as positive targets over less-preferred data as negatives, either using a reward model or direct optimization. Self-Play fine-tuning (Chen et al., 2024b; Yuan et al., 2024) further introduces iterative self-improvement, in which the model generates negative targets and compares them with the corresponding supervised targets that serve as the positives.

Despite their different origins—contrastive learning focuses on semantic similarity and preference optimization centers on human judgment—both approaches leverage comparative signals to distinguish desirable from undesirable outcomes. This common framework can be interpreted as maximizing mutual information between the data and the underlying labels (e.g., class or preference) (Tschannen et al., 2020; Oord et al., 2018; Xiao et al., 2025), which yields better performance than purely supervised learning.

While recent subject-driven TTI method (Miao et al., 2024) proposes reward models based on self-supervised training and applies per-subject preference optimization, it relies on reward models to estimate fidelity gaps and guide learning. In contrast, we emphasize the importance of negative signals in fine-tuning zero-shot subject-driven TTI models and automatically generate negative targets with fidelity gaps—without relying on any reward model—to facilitate comparison-based learning and enhance subject fidelity.

We further differentiate our method from a direct application of Diffusion-DPO (Wallace et al., 2024). Our motivation is specifically designed to solve the critical "mode-narrowing" challenge essential for subject-driven generation with negative targets while Diffusion-DPO aims to enhance general image quality. Moreover, Diffusion-DPO relies on the existing curated preference paired dataset, which are typically collected via human annotation, yet a simple extension of this for subject-driven generation is impractical as there is no such public dataset for the nuanced task of subject-fidelity, and manually collecting such pairs would be prohibitively costly. Therefore, our core methodological contribution is not the learning formulation itself, but rather the construction of effective negative targets and their subsequent use to enhance mode-narrowing ability.

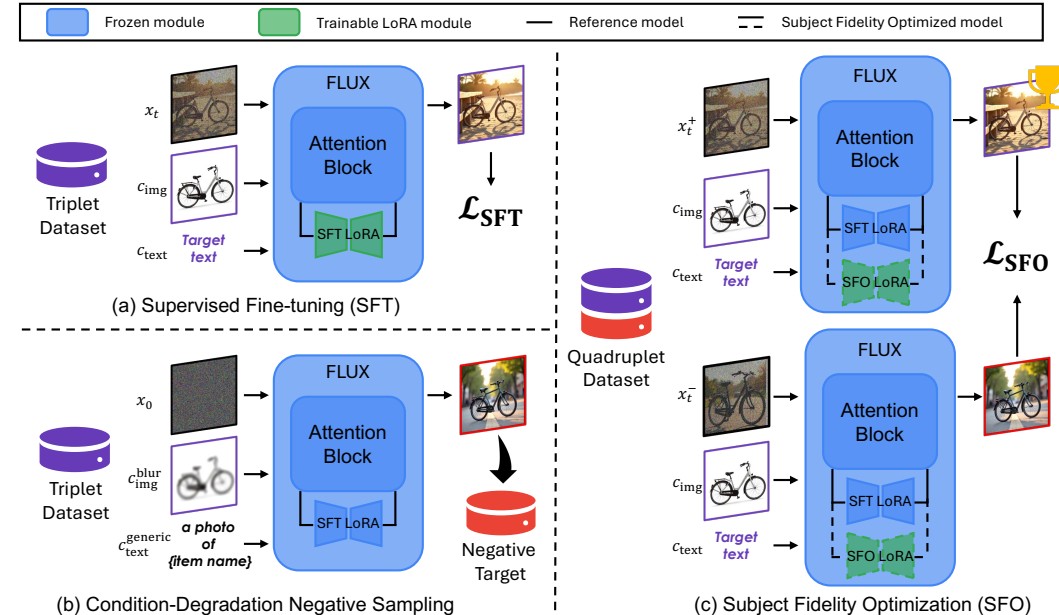

Figure 2: **Overall framework** (a) Previous supervised fine-tuning methods utilize the triplet dataset to generate the target image conditioned on a given reference image and target text prompt. (b) From a supervised fine-tuned model, we synthesize negative target data with CDNS, extending the triplet dataset into a quadruplet dataset containing informative negatives. (c) We further fine-tune the supervised fine-tuned model with a quadruplet dataset with SFO to distinguish positive and negative target data given the same condition.

## 3 METHOD

### 3.1 PROBLEM SETTING

We build upon the zero-shot subject-driven TTI framework, specifically the supervised fine-tuned transformer sequence extension method (Tan et al., 2024). This method fine-tunes the model on a triplet dataset (e.g., Subject200K (Tan et al., 2024)) composed of target image $x_{\text{tgt}}$, target text $c_{\text{text}}$, and reference image $c_{\text{img}}$ by incorporating latent tokens of reference images into the input sequence of MM-DiT (Esser et al., 2024) and using the same diffusion loss[1] formulation employed during the pre-training stage as shown in Fig. 2 (a),

$$\mathcal{L}_{\text{SFT}} = \mathbb{E}_{t \sim p(t), (x_{\text{tgt}}, c_{\text{text}}, c_{\text{img}}) \sim \mathcal{D}} \left[ \left\| f_\theta(x_t, t, c_{\text{text}}, c_{\text{img}}) - (\epsilon - x_{\text{tgt}}) \right\|^2 \right], \quad (1)$$

where $x_t = (1-t)x_{\text{tgt}} + t\epsilon$ is the interpolated image between the target image and random Gaussian noise $\epsilon \sim \mathcal{N}(0, I)$.

This triplet-based supervised fine-tuning implicitly guides the model to mimic the target data by leveraging the reference image as additional information. However, we observe that simply adding reference image conditioning and continuing training with the same loss is insufficient for subject-driven TTI, as it fails to sufficiently narrow the sampling toward a specific subject mode that reflects fine-grained details, and instead continues sampling subjects with irrelevant attributes. For example, the results for the poop emoji toy in the first row of Fig. 1 preserve global semantics such as object class but fail to capture the exact color or subject-specific details. We additionally validate this limitation with a simple toy experiment that attempts to narrow diffusion model sampling to a specific mode, the results of which are provided in Sec. A of the Appendix. This motivates a comparison-based fine-tuning by incorporating negative targets alongside original targets. It explicitly guides the model to favor the desired subject with fine-grained details while suppressing irrelevant ones.

---

[1]While our base model is a flow matching model, we refer to the vector field loss of the flow matching model as the diffusion loss to express generality.

## 3.2 SUBJECT FIDELITY OPTIMIZATION (SFO)

Building on the formulation of the pairwise comparison objective from direct preference optimization (Rafailov et al., 2023; Wallace et al., 2024; Chen et al., 2024b; Yuan et al., 2024), we propose a novel Subject Fidelity Optimization (SFO) strategy that focuses on enhancing subject fidelity with respect to a reference conditioning image in subject-driven text-to-image generation by leveraging tailored negative target image, as illustrated in Fig. 2(c). Given a quadruplet dataset that includes condition $c = (c_{\text{text}}, c_{\text{img}})$ along with a positive target image $x_{\text{tgt}}^+$ that has better subject fidelity and a negative target image $x_{\text{tgt}}^-$ that has worse subject fidelity, the following learning objective encourages the model to increase the probability of sampling the image with higher fidelity compared to the image with lower fidelity,

$$\mathcal{L} := -\mathbb{E}_{(x_{\text{tgt}}^+, x_{\text{tgt}}^-, c) \sim \mathcal{D}} \left[ \log \left( \sigma \Big( \beta \Big( \log \frac{p_\theta(x_{\text{tgt}}^+|c)}{p_{\text{ref}}(x_{\text{tgt}}^+|c)} - \log \frac{p_\theta(x_{\text{tgt}}^-|c)}{p_{\text{ref}}(x_{\text{tgt}}^-|c)} \Big) \Big) \right) \right], \tag{2}$$

where $p_\theta$ is the optimized model distribution, $p_{\text{ref}}$ is the reference distribution which is a supervised fine-tuned model distribution in our case, and $\beta$ is the regularization parameter. By leveraging the equivalence between minimizing the KL divergence (i.e., maximizing data likelihood) and the diffusion loss as its surrogate (Lipman et al., 2023; Silveri et al., 2024), the above objective can be approximated by the following SFO objective formulation:

$$\mathcal{L}_{\text{SFO}} = -\mathbb{E}_{t \sim p(t),\ (x_{\text{tgt}}^+, x_{\text{tgt}}^-, c) \sim \mathcal{D}} \left[ \log \sigma \Big( -\beta \Big( \Delta_\theta(x_{\text{tgt}}^+, x_{\text{tgt}}^-, t, c) - \Delta_{\text{ref}}(x_{\text{tgt}}^+, x_{\text{tgt}}^-, t, c) \Big) \Big) \right],$$

$$\Delta_*(x_{\text{tgt}}^+, x_{\text{tgt}}^- t, c) = \big\| f_*(x_t^+, t, c) - (\epsilon - x_{\text{tgt}}^+) \big\|^2 - \big\| f_*(x_t^-, t, c) - (\epsilon - x_{\text{tgt}}^-) \big\|^2, \tag{3}$$

where $p(t)$ is the uniform distribution $\mathcal{U}[0, 1]$, $x_t^+ = (1-t)x_{\text{tgt}}^+ + t\epsilon$, $x_t^- = (1-t)x_{\text{tgt}}^- + t\epsilon$ are interpolated images between each target image and same random noise $\epsilon \sim \mathcal{N}(0, I)$. In Sec. B and Sec. C of the Appendix, we provide a detailed derivation of this loss, as well as a mutual information–based justification that theoretically grounds its role in enhancing subject fidelity. We further confirm the effectiveness of comparison-based learning for fine-tuning diffusion models toward specific mode sampling through a toy experiment, the result of which is presented in Sec. A of the Appendix.

As SFO aims to enhance subject fidelity by capturing fine-grained details of the target subject, we focus on fine-tuning on timestep regions where such details begin to emerge during the generation process. Rather than blindly following the uniform timestep distribution $t \sim \mathcal{U}[0, 1]$ as used in theoretical formulations (Eq. 3), we adopt insights from prior works (Choi et al., 2022; Esser et al., 2024; Karras et al., 2022; Kingma & Gao, 2024) that emphasize the training in middle timesteps. Accordingly, we sample timesteps from a logit-normal distribution during the fine-tuning process, where $\text{logit}(t) = \log(\frac{t}{1-t}) \sim \mathcal{N}(0, I)$, to place more weight on this critical region.

## 3.3 CONDITION-DEGRADATION NEGATIVE SAMPLING (CDNS)

The core idea of SFO is to introduce the target pair $(x_{\text{tgt}}^+, x_{\text{tgt}}^-)$ and enhance subject-driven text-to-image generation through comparative signals between targets. Achieving this necessitates pairs that differ in subject fidelity under identical conditions; however, constructing such pairs is generally infeasible in practice and highly challenging. A promising alternative to construct such pairs is Self-Play (Chen et al., 2024b; Yuan et al., 2024), which performs preference optimization by treating the original targets in the supervised dataset as positives and pairing them with supervised fine-tuned model-generated data as negatives. Since Self-Play constructs target pairs by expanding a supervised dataset with a supervised fine-tuned model, it is a particularly practical option for subject-driven generation, where both a supervised dataset and a fine-tuned model are already available.

However, despite its superior performance over previous preference optimization methods (Rafailov et al., 2023; Wallace et al., 2024) in other tasks, its direct application to subject-driven text-to-image generation does not achieve satisfactory results. In particular, naïve Self-Play generates negatives by conditioning on the same inputs as the positives, often yielding nearly identical contexts and insufficient fidelity discrepancies (see row 3 of Fig. 3(a)). To address this limitation, we propose

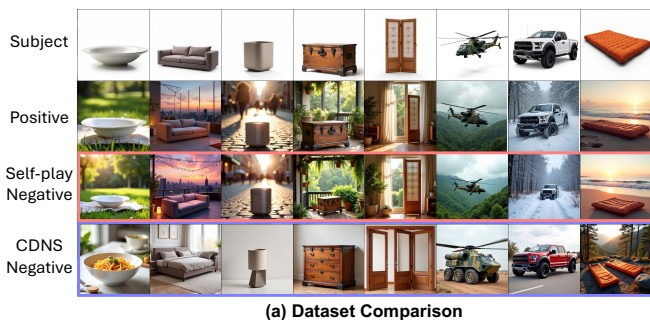 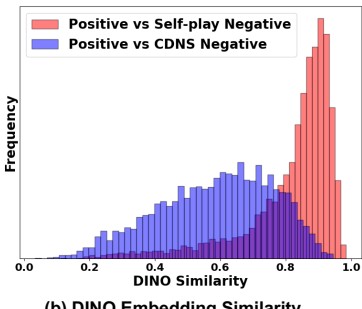

**(a) Dataset Comparison**  **(b) DINO Embedding Similarity**

Figure 3: **Dataset construction comparisons** (a) We present examples of synthesized negative targets from each naïve Self-Play method and our CDNS with given conditions. (b) While the negative targets of naïve Self-Play have high similarity with positive samples, our negative targets from CDNS demonstrate diverse pairwise gaps between targets and enable more effective optimization.

Condition-Degradation Negative Sampling (CDNS), in which supervised fine-tuned model synthesizes tailored negatives that provide more informative comparisons for subject-driven generation. CDNS deliberately applies degradation in both image and text conditioning modalities during negative data synthesis to induce further discrepancies in fine details with diverse levels of text alignments , as illustrated in Fig. 2(b): (1) We apply Gaussian blur to the conditioning image, denoted as $c_{\text{img}}^{\text{blur}}$, which removes subject-specific fine details. This prevents the synthesized negative target images from reflecting the fine-grained details of the target subject, thereby inducing a subject fidelity gap. (2) To diversify the text alignment of negative targets, we replace context-rich prompts $c_{\text{text}}$ with more generic and ambiguous phrases $c_{\text{text}}^{\text{generic}}$, as *"a photo of {item name}"*. This diversity in text alignments regularizes the model to preserve subject fidelity regardless of contextual variation.

Owing to the synergistic effects of degrading both the visual and textual conditions, CDNS facilitates the generation of negative targets that deviate in subject fidelity and exhibit broader context diversity. This is evidenced by the examples in row 4 of Fig. 3(a) and the shifted DINO similarity distribution between positive and negative targets in Fig. 3(b) compared to the Self-Play baseline. The distribution represents a wide spectrum of fidelity degradation—ranging from subtle to significant differences. These samples with diverse difficulty levels yield more effective and informative pairs for comparison-based learning, consequently providing robust learning signals. Detailed analysis regarding this diverse fidelity gap is provided in Sec. D.6 of the Appendix.

## 4 EXPERIMENTS

### 4.1 EXPERIMENTAL SETTINGS

**Implementation Details** Our SFO starts from OminiControl (Tan et al., 2024), which we retrained and hereafter refer to as the SFT-base. This model couples an SFT LoRA with pre-trained FLUX.1-Dev[2] (Labs, 2024) and is trained on $512 \times 512$ images from the Subject200K dataset following official implementation. For SFO, we append an additional rank-16 SFO LoRA module (Hu et al., 2022) to the SFT-base and optimize only its weights for computational efficiency. We set the hyperparameter $\beta = 1000$ and use the Prodigy optimizer (Mishchenko & Defazio, 2023). In negative target synthesis, we apply the Gaussian blur with radius 5 to degrade the conditioning image $c_{\text{img}}$ into $c_{\text{img}}^{\text{blur}}$. For both negative target and evaluation image generation, we sample for 28 steps using a classifier-free guidance scale of 3.5 (Ho & Salimans, 2022; Meng et al., 2023). More detailed information about the fine-tuning of SFO is provided in Sec. D.2 of the Appendix. While our main experiments utilize the most advanced, state-of-the-art text-to-image backbone model available, we also demonstrate the architectural generalizability of SFO by applying it to a baseline with a different backbone model (IP-Adapter based on SD-XL), as detailed in Sec. G of the Appendix.

**Evaluation Settings** We evaluate our method using DreamBench (Ruiz et al., 2023), a widely adopted benchmark for subject-driven TTI, which contains 30 unique subjects, each paired with

---

[2]FLUX.1-Dev: https://huggingface.co/black-forest-labs/FLUX.1-dev

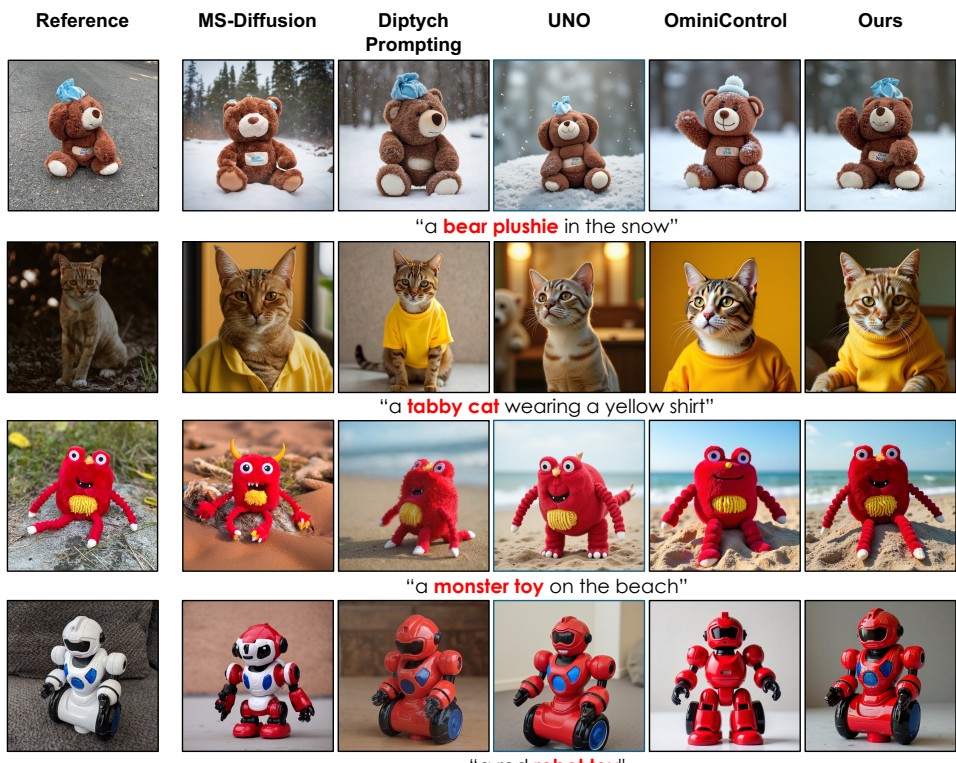

Figure 4: **Qualitative comparisons** Our method captures fine-grained details better than the baselines, such as the font on the abdomen of the bear plushie (row 1) or the limbs of the monster toy (row 3). Results in each row are generated with the same random seed for fairness.

25 target prompts. For each prompt, we generate 4 images using different seeds, resulting in 3,000 images in total for evaluation. We assess performance based on subject fidelity and text alignment, measured with both automatic metrics and human evaluations. For automatic evaluation, we compute the cosine similarity between the generated image and the reference image using embeddings from the DINO encoder (Caron et al., 2021) and the CLIP image encoder (Radford et al., 2021) to quantify subject fidelity. We also compute the cosine similarity between the CLIP image encoder embedding of the generated image and the CLIP text encoder (Radford et al., 2021) embedding of the target text prompt as a text alignment metric. For human evaluation, we conduct pairwise A/B testing via Amazon Mechanical Turk (AMT).

### 4.2 MAIN COMPARISONS

We compare our SFO against various zero-shot subject-driven TTI families: (1) encoder-based method, MS-Diffusion (Wang et al., 2025), (2) inpainting reinterpretation-based method, Diptych Prompting (Shin et al., 2025), and (3) transformer sequence extension methods, including Omini-Control (Tan et al., 2024), which is our base model fine-tuned with a supervised objective in Eq. 1, and recently proposed methods, UNO (Wu et al., 2025) and Kontext (Batifol et al., 2025).

**Qualitative Results** The qualitative comparison results are presented in Fig. 4. SFO preserves fine-grained subject details remarkably better than all baselines, including our SFT-base model, Omini-Control (Tan et al., 2024), which is trained on the same data but without CDNS negatives. This clearly highlights the advantage of our comparative learning, which enables more precise subject fidelity and regularizes alternative attributes compared to supervised fine-tuning. In contrast, MS-Diffusion (Wang et al., 2025) generates similar objects by reflecting the reference subject from a global perspective; yet, due to the CLIP image encoder's coarse semantic information, they fail to generate the exact same subject and often miss the context expressed in the text prompt. Diptych Prompting (Shin et al., 2025), the inpainting-based re-interpretation of subject-driven TTI, relies

Table 1: **Quantitative comparisons** We present the comparison results of our method against various baselines in automatic metrics. Higher values indicate better performance in all metrics. † indicates a re-tranied model by us with official implementation and we refer to it as SFT-base hereafter.

| Method | Model | DINO | CLIP-I | CLIP-T |
|---|---|---|---|---|
| Subject-Diff (Ma et al., 2024) | SD-v1.5 | 0.711 | 0.787 | 0.303 |
| MS-Diff (Wang et al., 2025) | SD-XL | 0.671 | 0.792 | 0.321 |
| IP-Adapter (Ye et al., 2023) | SD-XL | 0.613 | 0.810 | 0.292 |
| SuTi (Chen et al., 2023) | Imagen | 0.741 | 0.819 | 0.304 |
| JeDi (Zeng et al., 2024) | SD-v1.4 | 0.679 | 0.814 | 0.293 |
| DiptychPrompting (Shin et al., 2025) | FLUX | 0.689 | 0.758 | 0.344 |
| EasyControl (Zhang et al., 2025) | FLUX | 0.652 | 0.789 | 0.325 |
| UNO (Wu et al., 2025) | FLUX | 0.760 | 0.835 | 0.304 |
| Kontext (Batifol et al., 2025) | FLUX | 0.762 | 0.833 | 0.321 |
| OminiControl (Tan et al., 2024) † | FLUX | 0.652 | 0.795 | 0.329 |
| SFO | FLUX | 0.767 | 0.834 | 0.324 |

Table 2: **Human evaluation** We report the results of pairwise comparisons based on human perceptual preferences between SFO and the baselines.

| Method | Subject Fidelity (%) | | | Text Alignment (%) | | |
|---|---|---|---|---|---|---|
| | *win* | *tie* | *lose* | *win* | *tie* | *lose* |
| MS-Diff (Wang et al., 2025) | 71.2 | 7.7 | 21.1 | 69.1 | 15.6 | 15.3 |
| DiptychPrompting (Shin et al., 2025) | 60.8 | 6.6 | 32.6 | 61.5 | 17.6 | 20.9 |
| UNO (Wu et al., 2025) | 64.7 | 6.8 | 28.5 | 61.5 | 18.4 | 20.1 |
| Kontext (Batifol et al., 2025) | 56.0 | 9.0 | 35.0 | 51.8 | 20.1 | 28.1 |
| OminiControl (Tan et al., 2024) | 76.5 | 7.2 | 16.3 | 66.2 | 16.8 | 17.0 |

solely on the diptych property and the capability of pre-trained TTI model, failing to capture fine details in subjects. UNO (Wu et al., 2025), a recent transformer sequence extension method, often struggles to capture fine-grained details of the subject and to adequately reflect the textual context.

**Quantitative Results** As a quantitative comparison, we provide the results of automatic metrics in Tab. 1. We also include the results of several additional baselines for more comprehensive evaluation. Our method achieves consistently superior performance, outperforming both our SFT-base (OminiControl) and recent strong baselines in subject fidelity, while maintaining competitive text alignment, thus striking a balance across metrics, indicating a sweet spot by maintaining a balanced performance across all metrics. This verifies the necessity of incorporating negative targets and explicit negative signals to suppress undesirable features and enhance subject fidelity in zero-shot subject-driven TTI.

**Human Evaluation** To further demonstrate the superiority of SFO from a human perception perspective, we conduct a human evaluation as a pairwise comparison between our SFO and several strong baselines on two key objectives of subject-driven TTI: subject fidelity and text alignment. For each aspect, participants are asked to choose the preferred option between the outputs generated by the two models, and we include detailed information about the questionnaire in the Appendix. Using Amazon Mechanical Turk (AMT), we collect a total of 450 responses from 150 participants. As presented in Tab. 2, SFO is preferred over all baselines in both aspects ($p < 10^{-5}$ in the Wilcoxon signed rank test), which is well-aligned with quantitative and qualitative comparisons.

**Additional Benchmark** While DreamBench (Ruiz et al., 2023) is the most widely adopted and representative benchmark composed of real-world subjects, we evaluate with an additional benchmark, CustomConcept101 (Kumari et al., 2023) to demonstrate SFO's generalization to more challenging and diverse real-world examples. We compare our SFO against our SFT baseline, OminiControl, and the results are presented in Tab. 3. On this diverse and challenging benchmark, our method also significantly outperforms the SFT baseline regarding subject fidelity with comparable text alignment, demonstrating the effectiveness of our CDNS-guided comparative learning strategy. We present the qualitative comparisons in the Appendix.

Table 3: **Quantitative comparisons for CustomConcpet101** We provide additional quantitative comparisons for additional benchmark, CustomConcept101 (Kumari et al., 2023), to demonstrate generalizability of our SFO on more challenging and diverse real-world examples.

| Method | Model | DINO | CLIP-I | CLIP-T |
|---|---|---|---|---|
| SFT (OminiControl) (Tan et al., 2024) | FLUX | 0.521 | 0.726 | 0.308 |
| SFO | FLUX | 0.656 | 0.785 | 0.296 |

### 4.3 ABLATION STUDY

**Training Strategy** As a fair baseline, we also compare to an SFT-additional model, which further fine-tunes an additional LoRA module using the supervised loss only with positive targets (Eq. 1) under the same training settings as SFO. While this variant performs similarly to the SFT-base, as demonsrated in Tab. 4, our SFO shows a significant improvement, indicating that supervised fine-tuning with positive data alone is insufficient to enhance subject fidelity further, and explicitly suppressing undesirable features with negative targets is necessary.

Table 4: **Dataset construction ablation** We compare our CDNS with other dataset construction methods.

| Method | DINO | CLIP-I | CLIP-T |
|---|---|---|---|
| SFT-base | 0.652 | 0.795 | 0.329 |
| SFT-additional | 0.650 | 0.794 | 0.330 |
| DPO-DINO | 0.651 | 0.795 | 0.329 |
| Self-Play | 0.685 | 0.799 | 0.326 |
| CDNS | 0.767 | 0.834 | 0.324 |

**Target Pair Construction** We study the effectiveness of CDNS by comparing with two conventional strategies for constructing target pairs in Tab. 4. In DPO-DINO (Rafailov et al., 2023; Wallace et al., 2024), we construct target pairs by generating two images from the SFT-base model and labeling the one with the higher DINO similarity as the positive target and the other as the negative target. Performance remains similar to that of the SFT-base, as many synthesized pairs are nearly identical, yielding weak learning signals due to insufficient differentiation. In the Self-Play setting (Chen et al., 2024b; Yuan et al., 2024), the original Subject200K target is treated as the positive target, and the SFT-base model synthesizes negatives from the same condition as discussed in Sec. 3.3 and Fig. 3. Self-Play shows improvement over SFT-base, benefiting from occasional generation failures, which create subtle pairwise differences. In contrast, our CDNS intentionally degrades conditioning inputs to increase failure cases and amplify pairwise gaps. This results in diverse and informative target pairs, improving training effectiveness and outperforming all baselines.

**Degradation Design Choices in CDNS** We investigate design variations of degradation for both image and text conditions in negative target synthesis, and summarize their results in Tab. 5.

For image degradation, we further explore two alternatives to $c_{\text{img}}^{\text{blur}}$: Gaussian noise degradation, in which Gaussian noise is added to the conditioning image $c_{\text{img}}$, and Semantic degradation, in which the white portrait conditioning image $c_{\text{img}}$ is replaced with the target image $x_{\text{tgt}}$ placed in a rich background, thereby introducing semantic noise that makes it more difficult to focus on the target subject. For text degradation, in addition to the generic text used in our main setting, we also examine conditioning on hallucinated text generated by GPT-4o, denoted as GPT-4o-hallucinated, where the original conditioning text $c_{\text{text}}$ is intentionally perturbed by producing slightly incorrect captions of the target image using an LLM. The detailed query prompt employed for generating hallucinated captions with GPT-4o is provided in Sec. E of Appendix.

When applying text degradation alone, as expected, many synthesized negatives still retain fine details, thereby narrowing the fidelity gap and limiting the effectiveness of comparative learning. In the image-only setting, only blur degradation proves effective, as it removes fine details from the conditioning image and thus causes the loss of fine-grained details in the negative targets, while other degradations yield little improvement over our SFT-base model. Finally, the dual-modality setting, in which both image and text degradations are applied, produces synergistic effects and achieves the best performance overall, with blur degradation being particularly effective in synthesizing informative negatives that facilitate capturing fine-grained details in comparison-based learning.

Table 5: **Condition degradation design choices** We further validate CDNS by conducting ablations on various degradation strategies—such as blur, noise, and text perturbation—to evaluate their contributions to subject fidelity and text alignment.

| | Img Cond. | Text Cond. | DINO | CLIP-I | CLIP-T |
|---|---|---|---|---|---|
| Dual Degradation | Blur | Generic | 0.767 | 0.834 | 0.324 |
| | Blur | GPT-4o-hallucinated | 0.734 | 0.823 | 0.325 |
| | Gaussian Noise | Generic | 0.731 | 0.820 | 0.325 |
| | Semantic | Generic | 0.751 | 0.830 | 0.324 |
| Image-only Degradation | Blur | - | 0.733 | 0.826 | 0.325 |
| | Gaussian Noise | - | 0.657 | 0.799 | 0.326 |
| | Semantic | - | 0.663 | 0.795 | 0.327 |
| Text-only Degradation | - | Generic | 0.656 | 0.791 | 0.330 |
| | - | GPT-4o-hallucinated | 0.668 | 0.797 | 0.328 |

Table 6: **RL comparison** We compare our SFO formulation to online-RL algorithm.

| Method | DINO | CLIP-I | CLIP-T |
|---|---|---|---|
| SFT (Tan et al., 2024) | 0.652 | 0.795 | 0.329 |
| Reward + DDPO (Black et al., 2023) | 0.641 | 0.790 | 0.330 |
| SFO | 0.767 | 0.834 | 0.324 |

**Comparative Learning Formulation** One of our core contributions, CDNS, enables comparative learning for subject-driven generation, for which we employ the DPO formulation to realize. To validate the effectiveness of implementation, we compare our approach against a reinforcement learning (RL) strategy using the same dataset. For the RL baseline, we train a modified ImageReward (Xu et al., 2023) model from scratch to accept additional reference image condition and utilize the DDPO (Black et al., 2023) methodology to perform on-policy RL-based preference optimization. The results in Tab. 6 show that the online RL approach yields results inferior to the baseline. We attribute this to the inherent difficulty of training a reward model to discern fine-grained subject fidelity. Additionally, as in limitations of Self-Play, the generated samples in the online RL setting mostly receive high scores from the imperfect reward model. This results in sparse negative feedback, failing to provide a sufficient learning signal for mode-narrowing. Moreover, online RL is computationally intensive due to iterative sampling at every training steps. These findings confirm the effectiveness of our SFO formulation, and we believe advancing reward models and online RL algorithms for such fine-grained mode-narrowing tasks remains a challenging direction for future work.

## 5 CONCLUSIONS

In this paper, we propose Subject Fidelity Optimization (SFO) to address the limitations of existing zero-shot subject-driven TTI frameworks that rely on supervised fine-tuning. Our SFO extends the triplet dataset to a quadruplet dataset by incorporating synthesized negative targets, and leverages comparative signals to improve subject fidelity. We also prioritize the middle timesteps—where fine-grained details emerge—to enhance fine-tuning effectiveness. We further propose Condition-Degradation Negative Sampling (CDNS), a novel technique that systematically degrades conditions to create informative and distinguishable negatives in a practical and effective manner. Our method consistently outperforms existing zero-shot approaches across automatic metrics and human evaluations. Extensive ablation studies further demonstrate the effectiveness of each proposed component, thereby confirming their significant contributions to overall performance. We believe that our fine-tuning strategy can serve as a stepping stone for developing more advanced and efficient methods in subject-driven TTI.

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

## A  EFFECTIVENESS OF NEGATIVE TARGETS IN DIFFUSION FINE-TUNING

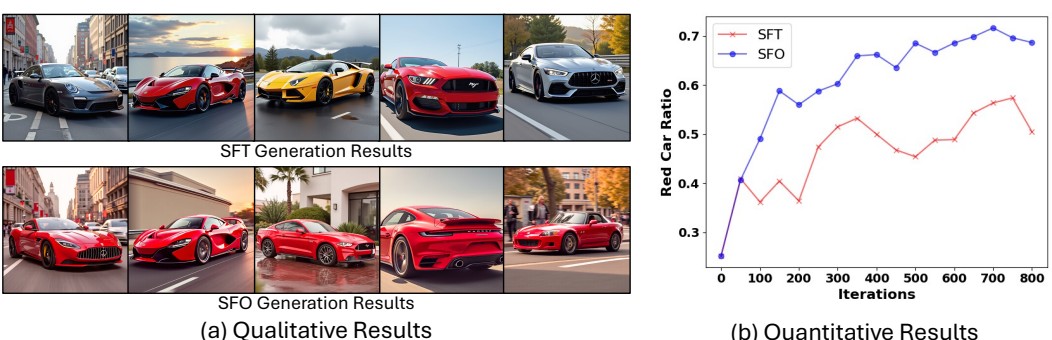

(a) Qualitative Results

(b) Quantitative Results

Figure A: **Toy experiment results** (a) Qualitative results: The top and bottom rows show images generated using the same random seed with each model, allowing for direct visual comparison. (b) Quantitative results: This plot shows the change in the proportion of images classified as red cars out of 100 generated samples, using a car color classifier.

To verify our motivation for introducing negative targets and highlight the insufficient learning signal in supervised fine-tuning, as discussed in Sec. 1 of the main paper, we conduct a toy experiment that emphasizes the role of negative targets in fine-tuning Text-to-Image (TTI) models. We design a simplified scenario that simulates narrowing the sampling distribution of a TTI model.

The original TTI model generates cars in various colors when given a generic text prompt $c_{\text{text}}$: "a photo of a car." We attempt to fine-tune this model to generate only red cars for the same text prompt $c_{\text{text}}$, and compare two fine-tuning approaches to ablate the effect of negative targets: (1) Supervised fine-tuning (SFT) using only red car images $x_{\text{red}}$ paired with the prompt $c_{\text{text}}$, and (2) Comparison-based fine-tuning (our SFO approach), using red car images $x_{\text{red}}$, the text prompt $c_{\text{text}}$, along with negative samples $x_{\text{various}}$ consisting of cars in various other colors.

In both cases, we attach a rank-16 LoRA module to FLUX.1-dev (Labs, 2024) and fine-tune only its weights until convergence (800 iterations), using 500 red car images and 500 various-colored car images. After fine-tuning, we generate 100 images using the prompt "a photo of a car" (identical to $c_{\text{text}}$ used during training), and evaluate the red car ratio, defined as the proportion of generated images classified as red cars using a car color classifier (TheDeveloperMask, 2020).

Qualitative and quantitative results are presented in Fig. A. Although the SFT model attempts to learn the red car distribution, it fails to sufficiently narrow the distribution over various car colors, still generating non-red cars and achieving a red car ratio of only up to 55%. In contrast, our SFO method explicitly leverages both red cars (as positives) and cars of other colors (as negatives), guiding the model to suppress non-red outputs. This leads to more successful red car generation, a higher red car ratio, and faster convergence. These experimental results validate both the necessity and advantage of incorporating a comparison-based learning signal for undesired content in the effective fine-tuning of TTI models.

## B  LOSS DERIVATION

Drawing inspiration from direct preference optimization (Rafailov et al., 2023), which encourages the model to favor the positive target over the negative one, we adopt a pairwise comparison objective for zero-shot subject-driven TTI. This objective is designed to distinguish positive targets with high subject fidelity from negative targets with lower fidelity.

We begin with the Bradley-Terry (BT) preference model (Bradley & Terry, 1952), which formulates the pairwise preference as a binary classification task between positive targets $x_{\text{tgt}}^{+}$ and negative targets $x_{\text{tgt}}^{-}$ :

$$\mathcal{L} := -\mathbb{E}_{(x_{\text{tgt}}^+, x_{\text{tgt}}^-, c) \sim \mathcal{D}} \left[ \log \frac{\exp(f_\phi(x_{\text{tgt}}^+, c))}{\exp(f_\phi(x_{\text{tgt}}^+, c)) + \exp(f_\phi(x_{\text{tgt}}^-, c))} \right] \tag{4}$$

$$= -\mathbb{E}_{(x_{\text{tgt}}^+, x_{\text{tgt}}^-, c) \sim \mathcal{D}} \left[ \log \sigma\Big( f_\phi(x_{\text{tgt}}^+, c) - f_\phi(x_{\text{tgt}}^-, c) \Big) \right], \tag{5}$$

where $c = (c_{\text{text}}, c_{\text{img}})$ represents the conditioning pair consisting of a text prompt and a reference subject image in subject-driven text-to-image generation, $\sigma(x)$ denotes the sigmoid function $\frac{1}{1+\exp(-x)}$, and $f_\phi$ is a learnable parametric function.

In our case, we implicitly define the parametric function as $f_\phi(x, c) = \beta \log \frac{p_\theta(x|c)}{p_{\text{ref}}(x|c)}$ which reflects the relative likelihood of the target model with respect to the reference model. Substituting this into the above, the equation becomes:

$$\mathcal{L} := -\mathbb{E}_{(x_{\text{tgt}}^+, x_{\text{tgt}}^-, c) \sim \mathcal{D}} \left[ \log \Big( \sigma\Big( \beta\Big( \log \frac{p_\theta(x_{\text{tgt}}^+|c)}{p_{\text{ref}}(x_{\text{tgt}}^+|c)} - \log \frac{p_\theta(x_{\text{tgt}}^-|c)}{p_{\text{ref}}(x_{\text{tgt}}^-|c)} \Big) \Big) \Big) \right], \tag{6}$$

where $\beta$ is a hyperparameter that controls the extent to which the optimized model deviates from the reference model.

However, flow-matching models (Lipman et al., 2023; Liu et al., 2023) cannot directly compute or model the log-probability $\log p_\theta(x)$. To address this, prior works (Lipman et al., 2023; Silveri et al., 2024) approximate the maximization of data likelihood or the minimization of KL divergence with a flow matching loss, which serves as a surrogate objective for optimization:

$$\min \ \mathcal{D}_{\text{KL}}(q(x)||p_\theta(x)) = \max \ \mathbb{E}_q[\log p_\theta(x)] \geq \mathbb{E}_q \left[ C - \underbrace{\int_0^1 \|f_\theta(x_t, t) - (\epsilon - x)\|_2^2 \ dt}_{\text{flow matching loss}} \right], \quad (7)$$

where $x_t = (1-t)x + t\epsilon$ denotes the linear interpolation between original image $x$ and Gaussian noise $\epsilon \sim \mathcal{N}(0, I)$, and $C$ is constant independent of $\theta$.

Based on this surrogate, we approximate the log-likelihood difference as:

$$\log \frac{p_\theta(x|c)}{p_{\text{ref}}(x|c)} = \log p_\theta(x|c) - \log p_{\text{ref}}(x|c) \tag{8}$$

$$\simeq \int_0^1 \left( \|f_\theta(x_t, t, c) - (\epsilon - x)\|_2^2 - \|f_{\text{ref}}(x_t, t, c) - (\epsilon - x)\|_2^2 \right) dt. \tag{9}$$

Consequently, our overall loss function is approximated as:

$$\mathcal{L} = -\mathbb{E}_{\mathcal{D}} \left[ \log \Big( \sigma\Big( \beta\Big( \log \frac{p_\theta(x_{\text{tgt}}^+|c)}{p_{\text{ref}}(x_{\text{tgt}}^+|c)} - \log \frac{p_\theta(x_{\text{tgt}}^-|c)}{p_{\text{ref}}(x_{\text{tgt}}^-|c)} \Big) \Big) \Big) \right] \tag{10}$$

$$\simeq -\mathbb{E}_{\mathcal{D}} \left[ \log \Big( \sigma\Big( \beta\Big( \int_0^1 \left\| f_\theta(x_t^+, t, c) - (\epsilon - x_{\text{tgt}}^+) \right\|_2^2 - \left\| f_{\text{ref}}(x_t^+, t, c) - (\epsilon - x_{\text{tgt}}^+) \right\|_2^2 dt \right.\right.\right.$$
$$\left.\left.\left. - \int_0^1 \left\| f_\theta(x_t^-, t, c) - (\epsilon - x_{\text{tgt}}^-) \right\|_2^2 - \left\| f_{\text{ref}}(x_t^-, t, c) - (\epsilon - x_{\text{tgt}}^-) \right\|_2^2 dt \Big) \Big) \Big) \right] \tag{11}$$

$$= -\mathbb{E}_{\mathcal{D}} \left[ \log \Big( \sigma\Big( \beta \int_0^1 \Delta_\theta(x_{\text{tgt}}^+, x_{\text{tgt}}^-, t, c) - \Delta_{\text{ref}}(x_{\text{tgt}}^+, x_{\text{tgt}}^-, t, c) \ dt \Big) \Big) \right], \tag{12}$$

where $\Delta_*(x_{\text{tgt}}^+, x_{\text{tgt}}^-, t, c) = \left\| f_*(x_t^+, t, c) - (\epsilon - x_{\text{tgt}}^+) \right\|^2 - \left\| f_*(x_t^-, t, c) - (\epsilon - x_{\text{tgt}}^-) \right\|^2,$

and we apply the Jensen's inequality, leveraging the convexity of the function $-\log \sigma(x)$,

$$\mathcal{L} = -\mathbb{E}_{\mathcal{D}} \left[ \log \left( \sigma \left( \beta \int_0^1 \Delta_\theta(x_{\text{tgt}}^+, x_{\text{tgt}}^-, t, c) - \Delta_{\text{ref}}(x_{\text{tgt}}^+, x_{\text{tgt}}^-, t, c) \, dt \right) \right) \right], \tag{13}$$

$$\leq -\mathbb{E}_{\mathcal{D}, t \sim p(t) = \mathcal{U}[0,1]} \left[ \log \left( \sigma \left( \beta \Delta_\theta(x_{\text{tgt}}^+, x_{\text{tgt}}^-, t, c) - \Delta_{\text{ref}}(x_{\text{tgt}}^+, x_{\text{tgt}}^-, t, c) \right) \right) \right], \tag{14}$$

$$= \mathcal{L}_{\text{SFO}} \tag{15}$$

in which $\int_0^1 \cdot dt$ is equivalent to expectation with uniform distribution $\mathcal{U}[0,1]$.

While the aforementioned SFO objective based on flow matching originally employs a uniform timestep distribution $p(t) = \mathcal{U}[0,1]$ across all steps, we empirically adjust this distribution $p(t)$ as $\text{logit}(t) = \log(\frac{t}{1-t}) \sim \mathcal{N}(0,1)$ to prioritize regions where fine subject details begin to emerge, as demonstrated in the ablation studies in Sec. 4.3 of the main paper and Sec. D.6.

## C  CONDITIONAL MUTUAL INFORMATION BETWEEN IMAGES AND SUBJECT FIDELITY

Contrastive Predictive Coding (CPC), also known as InfoNCE (Oord et al., 2018), is a widely used approach in self-supervised learning and is commonly employed to estimate mutual information. In particular, it has been shown that the conditional mutual information between a target variable $X$ and its implicit label $Y$, given a conditioning variable $C$, can be bounded by the InfoNCE loss (Ma et al., 2021; Tsai et al., 2022),

$$CMI(X; Y|C) \geq \text{InfoNCE} := \sup_f \sum_{i=1}^n \left[ \log \left( \frac{\exp(f(x_i, y_i))}{\exp(f(x_i, y_i)) + \sum_{j=1}^m \exp(f(x_j, y_j))} \right) \right], \tag{16}$$

where the positive pairs $(x_i, y_i)$ are sampled from the joint distribution $p(x, y|c)$, and negative pairs $(x_j, y_{j \neq i})$ are sampled from the product of marginals $p(x|c)p(y|c)$.

Following prior works (Oord et al., 2018; Tschannen et al., 2020; Xiao et al., 2025), we demonstrate that, in our case, the conditional mutual information between the target image $X$ and the implicit label $Y$—which denotes whether an image achieves high subject fidelity—given the conditioning variable $C$ can similarly be bounded as:

$$CMI(X; Y|C) \geq \log \left( \frac{\exp(f(x_i, y_i))}{\exp(f(x_i, y_i)) + \exp(f(x_j, y_j))} \right) \tag{17}$$

$$= \log \left( \frac{\exp(f(x^+, y=1))}{\exp(f(x^+, y=1)) + \exp(f(x^-, y=0))} \right) \tag{18}$$

$$= \log \left( \sigma \left( f(x^+, y=1) - f(x^-, y=0) \right) \right), \tag{19}$$

where $x^+$ denotes a positive target with high subject fidelity, labeled as $y = 1$, and $x^-$ denotes a negative target with low subject fidelity, labeled as $y = 0$, both under the same conditioning $c$.

This bound corresponds exactly (up to a sign) to the formulation of the Bradley-Terry preference model assumed in Sec. B. Therefore, our loss function can be interpreted as optimizing the model to maximize the conditional mutual information between the generated image and its subject fidelity with respect to the reference subject, given the condition. This provides a theoretical justification for the subject fidelity improvement achieved by our method.

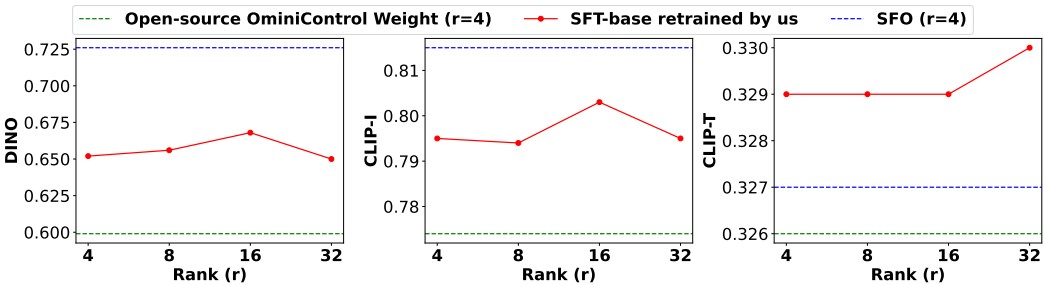

Figure B: **SFT-base rank ablation** We investigate various configurations of the SFT-base model prior to applying SFO and validate that training a LoRA module with rank 4 from scratch is sufficient for the SFT-base.

## D    SUBJECT-DRIVEN GENERATION

### D.1    SFT-BASE MODEL

Before fine-tuning with SFO, we re-train our SFT-base model, OminiControl (Tan et al., 2024), from scratch by following the official implementation, using a rank-4 LoRA (Hu et al., 2022) module (hereafter referred to as SFT LoRA). To validate the effectiveness of our base model, we compare it against the officially released open-source weights of OminiControl, as well as models with various SFT LoRA ranks. The entire results are presented in Fig. B. The open-source weights yield inferior performance compared to our re-trained model, and increasing the LoRA rank does not lead to any further improvement in subject-driven TTI performance. These results validate our decision to re-train the SFT LoRA with rank 4 as the SFT-base model, and highlight the necessity of introducing a new learning signal beyond further enhancing supervised fine-tuning.

### D.2    SFO IMPLEMENTATIONS

For SFO fine-tuning, we synthesize approximately 5,000 negative target data using the SFT-base model, OminiControl, re-trained by us from scratch in Sec. D.1 on a randomly selected subset of collection2 from the Subject200K dataset (Tan et al., 2024). This synthesis process takes about 9 seconds per image, totaling approximately 3 hours when running with a batch size of 1 on four NVIDIA L40 GPUs without data parallelism. Using this randomly sampled subset of Subject200K and the corresponding synthesized negative targets, we fine-tune our SFO starting from the SFT-base on four NVIDIA L40 GPUs with a batch size of $4$ for $300$ iterations without gradient accumulation. This procedure takes only 1.5 hours and yields substantial performance improvements.

Our PyTorch code implementation for calculating the SFO objective is presented in Fig. C.

### D.3    EVALUATION DETAILS

The DreamBench benchmark dataset (Ruiz et al., 2023) consists of 30 subjects, including 9 pets (cats and dogs) and 21 distinct objects (e.g., backpacks, sunglasses, characters). For each subject, the dataset provides 25 prompts designed to induce recontextualization, property modification, or accessorization. Following the setup in (Shin et al., 2025), we augment each subject name with descriptive keywords to improve subject fidelity in zero-shot methods. This augmentation strategy is consistently applied across all zero-shot baselines for fair comparison.

The following is a summary of our descriptive subject names in the form of (directory name in dataset, subject name):

- backpack, backpack
- backpack_dog, backpack
- bear_plushie, bear plushie
- berry_bowl, 'Bon appetit' bowl

```python
# Inputs
# positive (N, C, L), negative (N, C, L),
# text_cond (N, C, L_t), img_cond (N, C, L)

# sample timestep t
t = torch.nn.functional.sigmoid(torch.normal(0, 1, size=(N,)))

# concat as batch level and repeat conditions
x_0 = torch.cat((positive, negative))
t = t.repeat(2)
text_cond = text_cond.repeat(2, 1, 1)
img_cond = img_cond.repeat(2, 1, 1)

# flow matching diffuse
x_1 = torch.normal((N, C, L)).repeat(2, 1, 1)
x_t = (1-t) * x_0 + t * x_1

# reference model forward
model.set_adapters(["ref"])
ref_pred = model.forward(x_t, text_cond, img_cond, t)
ref_loss = ((ref_pred - (x_1 - x_0)) ** 2).mean([1, 2])
ref_loss_pos, ref_loss_neg = ref_loss.chunk(2, dim=0)
ref_diff = (ref_loss_pos - ref_loss_neg)

# SFO model forward
model.set_adapters(["ref", "sfo"])
model_pred = model.forward(x_t, text_cond, img_cond, t)
model_loss = ((model_pred - (x_1 - x_0)) ** 2).mean([1, 2])
model_loss_pos, model_loss_neg = model_loss.chunk(2, dim=0)
model_diff = (model_loss_pos - model_loss_neg)

# SFO loss
loss = - torch.nn.functional.logsigmoid(-beta * (model_diff -
    ref_diff))
```

Figure C: **PyTorch implementation of a SFO fine-tuning**

- can, 'Transatlantic IPA' can
- candle, jar candle
- cat, tabby cat
- cat2, grey cat
- clock, number '3' clock
- colorful_sneaker, colorful sneaker
- dog1, fluffy dog
- dog2, fluffy dog
- dog3, curly-haired dog
- dog5, long-haired dog
- dog6, puppy
- dog7, dog
- dog8, dog
- duck_toy, duck toy
- fancy_boot, fringed cream boot
- grey_sloth_plushie, grey sloth plushie

- monster_toy, monster toy
- pink_sunglasses, sunglasses
- poop_emoji, toy
- rc_car, toy
- red_cartoon, cartoon character
- robot_toy, robot toy
- shiny_sneaker, sneaker
- teapot, clay teapot
- vase, tall vase
- wolf_plushie, wolf plushie

## D.4 HUMAN EVALUATION DETAILS

Following prior studies (Ruiz et al., 2023; Shin et al., 2025), we conduct a human evaluation using an A/B test, in which participants compare the outputs of our SFO method against those of each baseline. In each evaluation instance, participants are shown two generated images—one from our SFO method and one from a baseline—alongside a reference subject image and a target text prompt. Participants are then asked to select the image that better satisfies each of the following objectives: subject fidelity, which measures the similarity between the subject in the generated image and the subject in the reference image; and text alignment, which measures how well the generated image reflects the target text prompt. To ensure reliability, we additionally include a quality control pair that compares an SFO-generated image against a random noise image. Participants who prefer the noise image over our SFO output are excluded from the analysis.

Our detailed instructions for questionnaire are as follows:

**Subject Fidelity**

- Given a reference image and two machine-generated images, select which machine-generated output better matches the subject of the reference image for each pair.
- Inspect the reference subject and then inspect the generated subjects. Select which of the two generated items reproduces the identity (item type and details) of the reference item. The subject might be wearing accessories (e.g. hats, outfits). These should not affect your answer. Do not take them into account. If you're not sure, select Cannot Determine / Both Equally.
- Given the provided subject in reference image, which machine-generated image best matches the subject of the reference image?

**Text Alignment**

- Given a reference image and two machine-generated images, select which machine-generated output better matches the target text for each pair.
- Inspect the target text and then inspect the generated items. Select which of the two generated items is best described by the target text. If you're not sure, select Cannot Determine / Both Equally.
- Given the provided target text, which machine-generated image is best described by the target text?

An example of our survey interface is included as a screenshot in Fig. D.

## D.5 PER-SUBJECT FINE-TUNING METHOD COMPARISONS

We compare our zero-shot method, SFO, with per-subject fine-tuning approaches: (1) Dream-Booth (Ruiz et al., 2023), a representative fine-tuning-based subject-driven TTI method, and (2) RPO (Miao et al., 2024), which employs a reward model defined as a harmonic function of a self-supervised model (ALIGN) similarity and directly optimize the TTI model for each reference subject

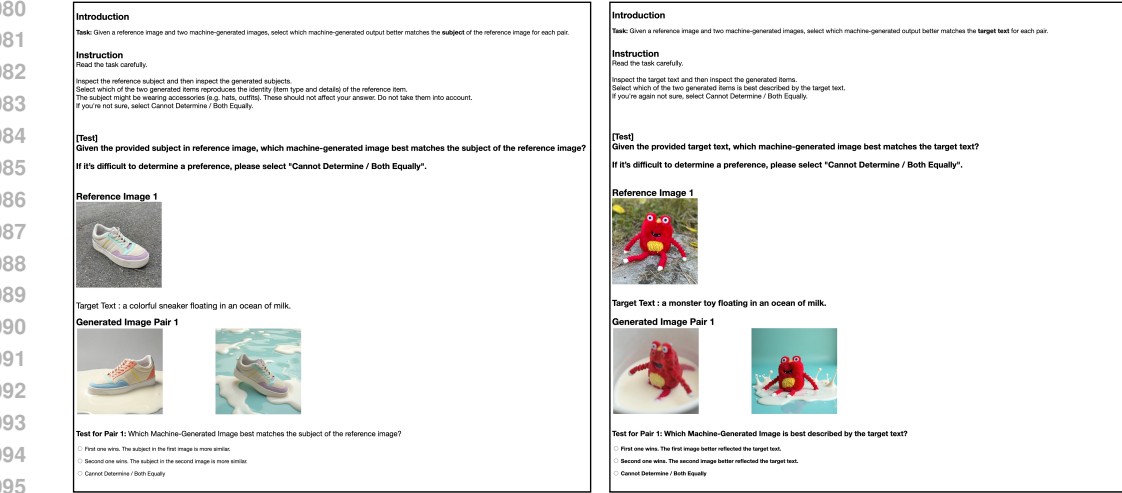

Figure D: **Amazon Mechanical Turk (AMT) survey interface example** We conduct a human evaluation using Amazon Mechanical Turk (AMT) via an A/B test, where participants select their preferred image from two generated results in terms of subject fidelity or text alignment.

Table A: **Comparisons to per-subject fine-tuning methods** Beyond the zero-shot subject-driven TTI methods, we compare our SFO to per-subject fine-tuning methods for comprehensive comparisons. Despite not requiring any fine-tuning on unseen subjects, SFO outperforms the fine-tuning-based approaches.

| Method | Model | DINO | CLIP-I | CLIP-T |
|---|---|---|---|---|
| RPO (rank= 32) | SD-v2.1 | 0.652 | 0.833 | 0.314 |
| RPO (rank= 32) | SD-XL | 0.682 | 0.800 | 0.340 |
| DreamBooth (rank= 32) | FLUX | 0.684 | 0.790 | 0.333 |
| SFO | FLUX | 0.767 | 0.834 | 0.324 |

using reward-based learning. We append a LoRA module with rank 32 to both methods and train only the LoRA weights for efficiency, with $\lambda_{val}$ as 0.5 for RPO. The results are presented in Tab. A. While both DreamBooth and RPO perform per-subject fine-tuning, SFO consistently outperforms both methods for novel subject in a zero-shot manner, demonstrating its superiority in terms of both efficiency and effectiveness.

## D.6 ADDITIONAL ABLATION

**Blur Degradation Sensitivity** We experimentally investigate the effect of varying degrees of blur degradation, which removes fine details from the conditioning image during CDNS synthesis, thereby inducing subject fidelity gaps between target pairs. Specifically, we vary the radius parameter $r \in \{2.5, 5.0, 10.0, 20.0\}$ of Gaussian blur to synthesize negative targets, and present the evaluation results for each configuration in Tab. B. We observe that strong blur ($r = 10$ or $r = 20$) removes excessive subject information from the conditioning image, yielding negatives that are overly easy to distinguish from the positives and consequently less useful for training, resulting in limited improvement. Conversely, weak blur ($r = 2.5$) results in conditions too similar to original conditioning image, still preserving much of the fine details and yielding only marginal improvements. Our results indicate that a moderate blur ($r = 5$) strikes the right balance: it effectively removes fine details while preserving the overall object shape, thereby producing more effective negative targets and achieving the best performance. This supports our motivation that introducing appropriate fine-detail discrepancies between target pairs effectively guides the model to capture aspects that SFO-trained models otherwise fail to reflect.

Table B: **Degradation sensitivity** We present the ablation results of blur degradation degree for synthesizing negative targets from CDNS in our SFO.

| $r$ | DINO | CLIP-I | CLIP-T |
|-----|------|--------|--------|
| 2.5 | 0.728 | 0.818 | 0.328 |
| 5.0 | 0.767 | 0.834 | 0.324 |
| 10.0 | 0.756 | 0.830 | 0.324 |
| 20.0 | 0.749 | 0.824 | 0.326 |

Table C: **Timestep reweighting in SFO** We present the ablation results of timestep distribution in our SFO.

| $p(t)$ | DINO | CLIP-I | CLIP-T |
|--------|------|--------|--------|
| $t \sim \mathcal{U}[0,1]$ | 0.713 | 0.814 | 0.328 |
| $\text{logit}(t) \sim \mathcal{N}(0,1)$ | 0.767 | 0.834 | 0.324 |
| $\text{logit}(t) \sim \mathcal{N}(-2,1)$ | 0.730 | 0.820 | 0.328 |
| $\text{logit}(t) \sim \mathcal{N}(2,1)$ | 0.664 | 0.796 | 0.331 |
| $\text{logit}(t) \sim \mathcal{N}(0,0.25)$ | 0.750 | 0.822 | 0.324 |

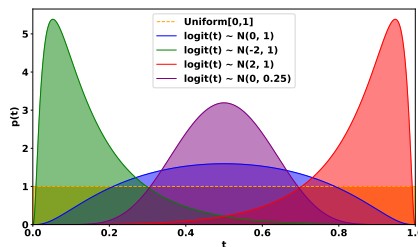

Figure E: **Timestep distribution**

**Timestep Ablation** We explore various timestep distributions to empirically identify which regions should be prioritized during SFO fine-tuning for improved subject fidelity. To this end, we replace the uniform distribution used in the theoretical formulation with a logit-normal distribution, varying its mean ($\mu$) and standard deviation ($\sigma$), as shown in Tab. C and Fig. E. While SFO with uniform timestep sampling already demonstrate superior performance compared to the SFT-base due to the introduction of negative data, we observe even better results when we focus fine-tuning on the middle timestep region using a logit-normal distribution with $\mu = 0$ and $\sigma = 1$. When the mean of the logit-normal distribution is increased (e.g., $\mu = 2$), fine-tuning is biased toward highly noised regions, where the noised positive image $x_t^+$ and the noised negative image $x_t^-$ become visually similar, making it difficult to distinguish subject fidelity. This hinders pairwise comparison learning, leading to inferior results. Using a logit-normal distribution with reduced mean of $\mu = -2$ shifts the sampling toward cleaner regions, which slightly weakens the training signal and yields lower performance than the centered case. Nevertheless, it still surpasses $\mu = 2$ and the uniform distribution, indicating that emphasizing earlier timesteps is generally more beneficial than focusing on highly noised or evenly distributed ones. Narrowing the fine-tuning region by reducing the standard deviation to $\sigma = 0.5$ leads to performance improvements by intensively training around the middle timesteps, but still shows slightly inferior performance compared to our default logit-normal distribution.

**Beta Ablation** We analyze the impact of the hyperparameter $\beta$ in the comparison-based learning objective of our SFO. In the loss formulation, $\beta$ serves to regularize deviation from the reference model: a larger value strongly constrains the adaptation of the optimized model, while a smaller value permits greater deviation. We evaluate $\beta \in \{500, 1000, 2000\}$, and the results are presented in Tab. D. Although all settings already yield strong performance, we select $\beta = 1000$ as our default setting since it achieves the best overall results.

**SFO LoRA Capacity** In Fig. F, we investigate how SFO's performance varies with model capacity by varying the LoRA rank. SFO exhibits robust performance with respect to rank variation, and we adopt rank 16 as our base setting as it balances expressiveness and efficiency. Notably, SFT-additional shows little performance change across different ranks, indiciating that positive data alone cannot introduce additional meaningful signal for training. Self-Play shows limited impact across different ranks due to the insufficient number of target pairs with meaningful differences. In contrast, SFO with CDNS consistently outperforms others across all ranks, showing steady improvements in subject fidelity without significant losses in text alignment, owing to the abundance of meaningful negatives. This highlights the effectiveness of comparative learning in SFO with CDNS-

Table D: **Beta ablation** We assess the model performance with varying hyperparameter $\beta$.

| $\beta$ | DINO | CLIP-I | CLIP-T |
|---|---|---|---|
| 500 | 0.755 | 0.818 | 0.323 |
| 1000 | 0.767 | 0.834 | 0.324 |
| 2000 | 0.725 | 0.817 | 0.327 |

Table E: **Additional LoRA** Directly fine-tuning the SFT LoRA module with SFO objective requires significant longer training for improvement.

| Method | DINO | CLIP-I | CLIP-T |
|---|---|---|---|
| SFT-base | 0.652 | 0.795 | 0.329 |
| + direct SFO | 0.652 | 0.795 | 0.329 |
| SFO | 0.767 | 0.834 | 0.324 |

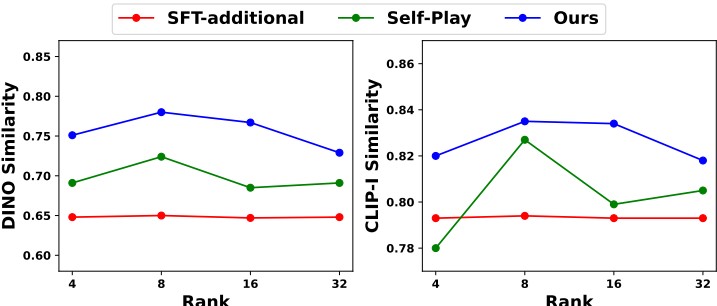

Figure F: **Ablation on SFO LoRA Rank** We investigate the effect of model capacity by varying the LoRA rank in fine-tuning stage. Unlike SFT-additional and Self-Play, SFO with CDNS consistently improves subject fidelity without harming text alignment.

generated negative targets, demonstrating its benefits that extend beyond model capacity when compared to simple supervised fine-tuning or naïve Self-Play.

**Additional LoRA** For more fair comparison, we conduct an ablation study which directly fine-tuning the SFT LoRA with the SFO objective, resulting in no performance improvement, as shown in Tab. E. This implies that our SFO fine-tuning complements SFT—which implicitly mimics the target data by referencing the subject image—and benefits from an additional LoRA module for effective knowledge integration. Using a separate LoRA module for SFO also allows an additional advantage, which provides independent control over model capacity only for subject fidelity optimization.

**Fidelity Degradation Diversity of Negatives from CDNS** To empirically validate the benefit of the diverse fidelity degradation levels produced by CDNS, we expand the CDNS generation to create more target pairs and categorize the pairs into 'High', 'Mid', and 'Low' sets based on their DINO similarity between target pair. To ensure a fair comparison, we equalize the total number of training samples across all settings with our original setting using full spectrum ('High+Mid+Low'). The results are presented in Tab. F. While all settings outperform the SFT baseline, the model trained solely on the 'Mid' set shows the largest improvement among individual settings, indicating that moderately difficult negatives are most informative in isolation compared to excessive or subtle discrepancy. However, the best performance is achieved when utilizing the full spectrum of diversity, demonstrating that exposing the model to a comprehensive range of difficulties contributes to more robust and effective training process.

Table F: **Ablation on the diversity of fidelity degradation in CDNS negatives** We present the ablation results of subject fidelity diversity among negative targets and demonstrate that the full spectrum of diversity from CDNS contributes to more effective training.

| Method | DINO | CLIP-I | CLIP-T |
|---|---|---|---|
| SFT | 0.652 | 0.795 | 0.329 |
| Low | 0.723 | 0.815 | 0.328 |
| Mid | 0.741 | 0.815 | 0.327 |
| High | 0.723 | 0.811 | 0.329 |
| High+Mid+Low | 0.767 | 0.834 | 0.324 |

## E    TEXT DEGRADATION USING GPT-4O HALLUCINATION

---
**Query Prompt for Generating Hallucinated Text with GPT-4o**

Please provide a short English description of this object photo. Use the following examples for generating description: {original conditioning text}. Include intentionally incorrect details (e.g., wrong color, shape) of object.

---

## F    ADDITIONAL SAMPLES

We include additional qualitative results of our SFO in Fig. G for diverse subjects and prompts. These results are representative samples demonstrating SFO's robust subject fidelity, and the overall performance is consistent with the strong quantitative metrics and the clear preference for our method observed in human evaluations.

While the vast majority of cases show excellent fidelity, we acknowledge minor failure cases, such as the slight degradation of fine text on the "Transatlantic IPA can" (row 1 of Fig. G). We attribute this artifact not to a fundamental limitation of our SFO learning strategy, but to other primary inherent constraints of the experimental setup: The backbone model's VAE compresses high-resolution images into a latent space. This compression can naturally cause information loss for extremely fine-grained details, such as small text.

This type of artifact is commonly observed in latent diffusion models . We are confident that this limitation can be readily addressed by future advancements, such as the adoption of improved VAE architectures or fine-tuning on higher-resolution datasets.

We also include qualitative comparisons on the challenging and more realistic subjects of Custom-Concept101 (Kumari et al., 2023), demonstrating improvements over our SFT baseline, in Fig. H.

## G    ARCHITECTURE GENERALIZATION

Beyond the single base backbone text-to-image model, we demonstrate the architecture-agnostic effectiveness of our SFO by applying it to the other supervised fine-tuned model based on different backbone–IP-Adapter (Ye et al., 2023) on U-Net-based Stable Diffusion XL (SD-XL) (Podell et al., 2024). The results are provided in Tab. G, which proves the improved performance in both subject fidelity and text alignment.

Table G: **Architecture generalization** We demonstrate the applicapability of SFO on other text-to-image diffusion backbones (SD-XL).

| Method | Model | DINO | CLIP-I | CLIP-T |
|---|---|---|---|---|
| IP-Adapter (Ye et al., 2023) | SD-XL | 0.613 | 0.810 | 0.292 |
| IP-Adapter + SFO | SD-XL | 0.664 | 0.833 | 0.308 |

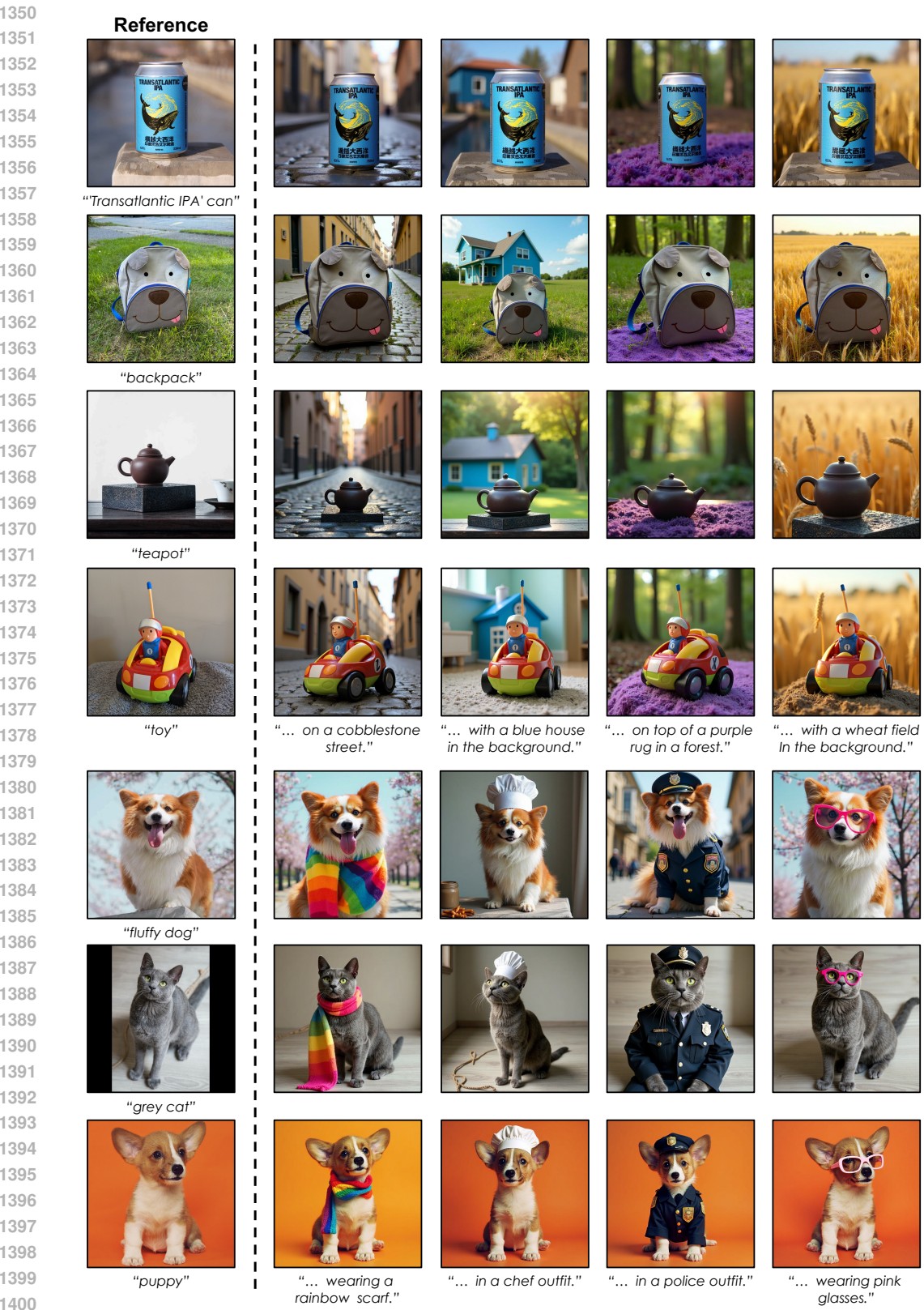

Figure G: **Additional samples** We provide additional qualitative results for various subjects and prompts. Zooming in enables a more detailed view.

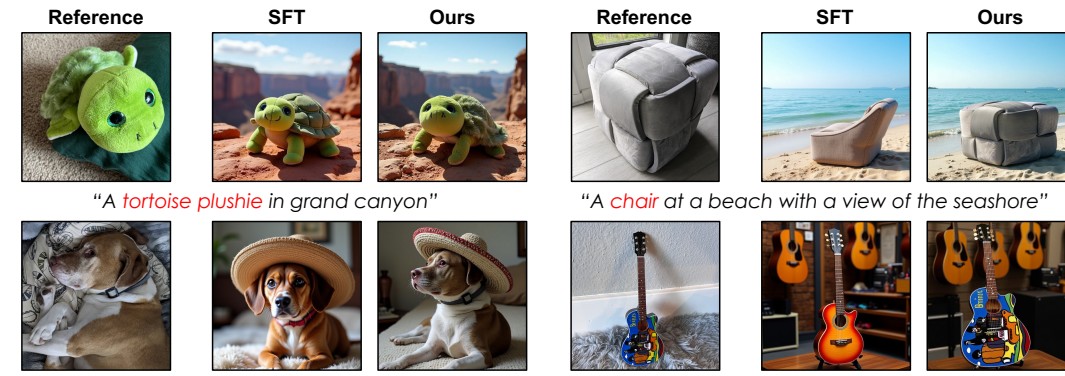

| Reference | SFT | Ours | Reference | SFT | Ours |

*"A tortoise plushie in grand canyon"*  *"A chair at a beach with a view of the seashore"*

*"A dog is wearing a sombrero"*  *"A guitar at display in a music shop."*

Figure H: **CustomConcept101 comparisons** We provide additional qualitative comparisons on diverse real-world subjects and prompts from CustomConcept101 (Kumari et al., 2023). Zooming in enables a more detailed view.

## H   PROMPTS IN TEASER IMAGE

**Prompts**

- *" Product photography, {item name} placed on a white marble table, white curtains, full of intricate details, realistic, minimalist, layered gestures in a bright and concise atmosphere, minimalist style "*
- *" a {item name} on the shelf at walmart on sale "*
- *" a large {item name} on the moon "*

**Item Names**

- toy
- yellow clock
- cartoon character
- wolf plushie

## I   LLM USAGE

For this work, we utilized Large Language Model, OpenAI's ChatGPT-5 (www.chatgpt.com), exclusively for writing-related tasks such as editing and formatting. The LLM was not involved in any aspect requiring a formal declaration, including the research design, methodological development, or in ensuring scientific rigor and originality. The authors take full responsibility for all content in this paper.

