# OpenReview forum: "Negative-Guided Subject Fidelity Optimization for Zero-Shot Subject-Driven Generation"
_ICLR.cc/2026/Conference — Submitted to ICLR 2026_

### Official Review · Reviewer_ajgK · 2025-10-25

**Soundness:** 2
**Presentation:** 3
**Contribution:** 2
**Rating:** 4
**Confidence:** 5

**Summary:**

This paper presents a method for enhancing the subject fidelity of the subject-driven image generation task via a negative-target-augmented strategy. The approach builds upon a diffusion-DPO-style algorithm and techniques for generating negative cases. Extensive experiments are carried out to prove the authors' motivation.

**Strengths:**

- The paper is clearly formatted and well-organized, making it easy for readers to follow the methodology, experimental setup, and results. The exposition is concise and accessible.
- The use of negative targets as auxiliary supervision during training is a sensible design choice. By explicitly modeling undesirable outputs, the method has the potential to avoid common failure modes in fine-tuned generation.

**Weaknesses:**

- The proposed framework closely follows the formulation of Diffusion-DPO and ultimately operates within the now-common SFT+DPO paradigm. As such, the methodological novelty is limited, and the contribution seems incremental.
- As shown in Figure G in Appendix, several generated samples exhibit noticeable fidelity degradation. For instance, distorted text on "can" and malformed shapes of "tea pot".
- All visual results are drawn from the Dreambooth benchmark. The exclusive reliance on this narrow domain raises concerns about the model’s generalization to more diverse or complex real-world inputs.

**Questions:**

- How does the method perform on inputs outside the Dreambooth setting, particularly on uncurated, real-world images?
- From a foundational perspective, Diffusion-DPO is inspired by reinforcement learning through self-exploration, where both win and lose cases should ideally be generated by the policy (i.e. the model) itself. However, like many recent works, this paper uses ground-truth images as the win case, which deviates from the original principle. Has the authors conducted analysis on the implications of this design choice?

---

> ### Author Response · Authors · 2025-11-21
> **Rebuttal by Authors**
>
> We thank you once again for dedicating your time to review our manuscript.
>
> We are very pleased that our dedicated efforts in the paper's writing and clarity have been recognized and especially grateful for your acknowledgment of our design choice of explicitly modeling undesirable outputs to avoid common failure modes.
>
> We will now address the concerns you have raised regarding our paper.
>
> > ### **[W1] … closely follows the formulation of Diffusion-DPO and ultimately operates within the now-common SFT+DPO paradigm …**
>
> While our method follows the DPO formulation to implement comparative learning, we clarify our contributions go beyond a direct application, centered on a different motivation and a novel, non-trivial data construction method applying for subject-driven generation. We further emphasize that the formulation alone is insufficient, as evidenced by our dataset construction ablation study; our negative targets synthesized from CDNS is a prerequisite for unlocking the potential of this formulation.
>
>
> **1. Difference in Motivation and Objective.**
>
> While existing Diffusion-DPO work focuses on improving general image quality, our primary objective is fundamentally different: we **aim to enhance the model's "mode-narrowing" ability**.
> We argue that for tasks like subject-driven generation—which require sampling from a very specific mode (i.e., matching the subject)—simple supervised fine-tuning (SFT) is insufficient. This is further demonstrated in our toy experiment that SFT cannot narrow the sampling distribution down sufficiently in Sec. A of our supplementary materials. We identified that **a comparative learning signal, driven by negative targets, is essential to enhance this ability** on both simple diffusion model sampling in toy experiments and complex subject-driven generation in the main paper, and we merely "borrow" the DPO formulation as an effective implementation of this novel learning strategy. To the best of our knowledge, we are the first to explicitly analyze and emphasize the role of negative targets for fine-tuning zero-shot subject-driven generation models.
>
> **2. The Non-trivial Extension due to Data Collection.**
>
> We further emphasize that **a "trivial extension" of Diffusion-DPO to our task is nontrivial in practice**. Diffusion-DPO typically learns from existing curated datasets with manually human-annotated pair-wise preference labels for general image quality. In contrast, for subject-driven generation, no such public dataset exists. The task **requires labels reflecting nuanced differences in subject fine-detail fidelity with respect to reference subjects, which are exceptionally difficult and prohibitively challenging to collect manually**. (It is worth noting that even collecting a Supervised Fine-Tuning (SFT) dataset is exceptionally challenging, as it requires gathering triplets of a subject image, a target text, and a corresponding target image. This difficulty is precisely why effective triplet dataset construction methods have emerged only very recently—lagging significantly behind the rapid advancements in text-to-image models—and why SFT itself has only just begun to demonstrate its full potential for this task.) Therefore, meaningful negative target construction becomes a critical and central component to implement our comparison-based fine-tuning. Our extensive ablation studies on dataset construction and their resulting performance differences validate this point.
>
> **3. Our Novel Contribution: CDNS.**
>
> This is precisely where **our key methodological contribution, CDNS, comes in**. CDNS is a novel and task-specific method for **automatically synthesizing target pairs with nuanced differences** in both subject fidelity and context from supervised fine-tuning dataset. It is what enables the successful application of the comparison-based learning formulation to the subject-driven generation task without the need for any manual labeling. Moreover, the resulting performance improvement varies drastically depending on the data construction strategy even when using the same comparative learning formulation, as supported by the table below (reorganized from Tab. 4 in the main paper) which presents the results of DPO formulation with various data construction strategies.
>
> |           | DINO  | CLIP-I | CLIP-T |
> |-----------|-------|--------|--------|
> | DPO-DINO  | 0.651 | 0.795  | 0.329  |
> | Self-Play | 0.685 | 0.799  | 0.326  |
> | CDNS      | 0.767 | 0.834  | 0.324  |
>
> This demonstrates that **simple DPO extension failed and the key element of our success is not the learning formulation itself, but rather the construction of effective negative targets via CDNS and their subsequent use in training.**
>
> We clarify this content more clearly in Sec. 2.2 of the main paper in the revised manuscript.

---

> ### Author Response · Authors · 2025-11-21
> **Rebuttal by Authors (2)**
>
> > ### **[W2] … several generated samples exhibit noticeable fidelity degradation. …**
>
> We sincerely thank the reviewer for this detailed and careful observation.
>
> We would like to respectfully highlight that our qualitative results, which were selected to be representative and not "cherry-picked," demonstrate excellent subject fidelity in the vast majority of cases, outperforming all baselines. While a few minor failure cases are naturally present, **the overall positive trend—and not a few select outliers—is clear and consistent with the strong preference for our method observed in the human evaluation**.
> Regarding the specific failure cases you pointed out:
>
> * On the "can" artifact: We acknowledge the minor degradation of small text on the "can." We will add this as a limitation in our revised manuscript. However, we note that this is not a fundamental limitation of our learning method, but rather an inherent consequence of the backbone model's VAE, which compresses the image into the latent space. The compression naturally causes some information loss, especially for such fine details. This artifact is consistently observed across other methods using the same latent diffusion backbone and resolution settings, and we believe it will be naturally resolved with future advancements in VAE architectures.
>
> * On the "teapot" artifact: We also acknowledge the minor artifact in one specific instance. However, we have observed that our method consistently generates high-fidelity results for this subject across numerous other examples and random seeds.
>
> We address the limitation you raised in greater detail in Sec. F of the Appendix.
>
> > ### **[W3] … the model’s generalization to more diverse or complex real-world inputs …**
> > ### **[Q1] … outside the Dreambooth setting, particularly on uncurated, real-world images?**
>
> We appreciate for raising this important concern regarding generalization to more diverse and complex subjects.
>
> We first would like to note that DreamBench is the most widely adopted and representative benchmark composed of sufficiently complex real-world subjects–serving as the primary and often sole benchmark for single subject-driven generation in most related literature.
>
> However, we agree with the reviewers' concern, and we **re-evaluate our method and our SFT baseline, OminiControl, on the CustomConcept101 dataset [1]**, which is more challenging and realistic than DreamBench, spanning diverse real-world subjects. For each concept, there are 20 prompts per concept that require background changes, insertion of new objects, style variations, or material/property changes of the subject. The results are as follows:
>
> |                    | DINO  | CLIP-I | CLIP-T |
> |--------------------|-------|--------|--------|
> | SFT (OminiControl) | 0.521 | 0.726  | 0.308  |
> | SFO                | 0.656 | 0.785  | 0.296  |
>
> On this diverse and challenging benchmark, our method also **significantly outperforms the SFT baseline regarding subject fidelity with comparable text alignment**, demonstrating the effectiveness of our CDNS-guided comparative learning strategy.
>
> We add this additional experimental results in Sec. 4.2 and Tab. 3 of the revised main paper, and Fig. H of the Appendix.
>
> ### **References**
>
> [1] Kumari et al., Multi-Concept Customization of Text-to-Image Diffusion, CVPR 2023.

---

> ### Author Response · Authors · 2025-11-21
> **Rebuttal by Authors (3)**
>
> > ### **[Q2] From a foundational perspective, Diffusion-DPO is inspired by reinforcement learning through self-exploration, where both win and lose cases should ideally be generated by the policy (i.e. the model) itself. …**
>
> Thank you for this insightful question regarding the foundational principles of our method.
>
> First, we would like to **respectfully clarify the training data of the original Diffusion-DPO**. Diffusion-DPO also trained SD-XL or SD-v1.5 on the Pick-a-Pic dataset [2], which is a collection of images generated by various models (including SD-XL beta, SD-v2.1, and DreamLike), not exclusively by the policy model being trained. Consequently, while theoretically inspired by self-exploration, **in practice, Diffusion-DPO operates more as an off-policy framework**. This approach of using pre-existing, non-policy data is also observed in other DPO-style methods like SPIN-Diffusion [3], which supports our design choice.
>
> That said, we still share the reviewer’s curiosity regarding the implications of using pure self-generated pairs, and we **did conduct a comparison of this exact design choice**, as presented in our dataset construction ablation (DPO-DINO in Tab. 3 of main paper). In that experiment, we used the SFT model (the policy) to generate image pairs for the same condition, then used DINO similarity as a proxy for preference labeling. We observed that the resulting performance improvement was insufficient. We found this was because the generated pairs often did not exhibit a significant fidelity gap, leading to a weak or non-existent learning signal. This very observation became the core motivation to make more distinct learning signals. Thus, we **prioritized the effectiveness of the learning signal over strict adherence to theoretical on-policy generation**, leading to our proposed CDNS framework.
>
> ### **References**
>
> [2] Kirstain et al., Pick-a-Pic: An Open Dataset of User Preferences for Text-to-Image Generation, NeurIPS 2023
>
> [3] Yuan et al., Self-Play Fine-Tuning of Diffusion Models for Text-to-Image Generation, ECCV 2024

---

### Official Review · Reviewer_3v7S · 2025-10-27

**Soundness:** 3
**Presentation:** 2
**Contribution:** 2
**Rating:** 4
**Confidence:** 4

**Summary:**

This paper proposes Subject Fidelity Optimization (SFO), a comparative learning framework for zero-shot subject-driven text-to-image generation. By introducing synthetic negative targets through a Condition-Degradation Negative Sampling (CDNS) strategy and reweighting diffusion timesteps, SFO explicitly guides the model to prefer high-fidelity subjects over degraded ones. Experiments show that SFO significantly improves subject fidelity and text alignment compared to existing supervised fine-tuning methods.

**Strengths:**

- **Comparative fine-tuning formulation:** The paper formalizes a pairwise optimization framework (SFO) that introduces explicit comparison between positive and negative targets, aligning with recent trends in preference-based optimization for diffusion models.

- **Automatic negative construction:** The proposed CDNS procedure provides a simple yet systematic way to synthesize negative samples by degrading visual and textual conditions, avoiding the need for manual annotation.

- **Comprehensive experiments:** The authors conduct quantitative, qualitative, and ablation analyses on standard benchmarks (e.g., DreamBench) under the FLUX backbone, illustrating the design choices and providing empirical observations of the method’s behavior.

**Weaknesses:**

**Lack of Novelty:** The proposed SFO framework lacks clear novelty, as applying DPO-style preference optimization to diffusion models has already been explored in prior works such as Wallace (2024) and related studies. The current formulation does not introduce fundamentally new theoretical insights or algorithmic mechanisms beyond these existing diffusion-based DPO approaches. The paper should clarify how SFO differs from and improves upon these earlier methods.

**Limited Performance:** Although the proposed SFO slightly improves DINO scores, the gain is marginal and not substantial enough to demonstrate a clear advantage in subject fidelity. Moreover, its CLIP-I and CLIP-T results remain below the state-of-the-art, suggesting that SFO provides limited overall improvement and may still struggle to balance subject fidelity with semantic and text–image alignment.

**Lack of Generalization Evidence:** The evaluation is limited to a single base model (FLUX) and one benchmark, making it unclear whether SFO generalizes to other diffusion backbones (e.g., SD-XL, Imagen) or broader subject-driven generation settings. Without cross-model validation, the claimed effectiveness of SFO may be dataset- or architecture-specific.

**Questions:**

- **Q1.** The paper shows that the negative samples generated by CDNS exhibit lower DINO similarity compared to those produced by the self-play method. Why does lower DINO similarity lead to better performance?

- **Q2.** What are the main differences between SFO and previous works such as Wallace (2024) and related studies?

---

> ### Author Response · Authors · 2025-11-21
> **Rebuttal by Authors**
>
> We extend our sincere gratitude for dedicating your time and effort to reviewing our manuscript.
>
> First and foremost, we are delighted that you recognized our key contributions: the design of a comparison-based learning framework, the automated negative synthesis method, and our comprehensive experimental validation.
>
> We will now provide a more detailed explanation to address the concerns you have raised.
>
> > ### **[W1] … lacks clear novelty, as applying DPO-style preference optimization to diffusion models …**
> > ### **[Q2] … main differences between SFO and previous works such as Wallace (2024) …**
>
> While DPO-style preference optimization is explored in Diffusion-DPO for general image generation quality, we clarify our contributions go beyond a direct application, centered on a different motivation and a novel, non-trivial data construction method applying for subject-driven generation. We further emphasize that the formulation alone is insufficient, as evidenced by our dataset construction ablation study; our negative targets synthesized from CDNS is a prerequisite for unlocking the potential of this formulation.
>
> **1. Difference in Motivation and Objective.**
>
> While existing Diffusion-DPO work focuses on improving general image quality, our primary objective is fundamentally different: we **aim to enhance the model's "mode-narrowing" ability**.
> We argue that for tasks like subject-driven generation—which require sampling from a very specific mode (i.e., matching the subject)—simple supervised fine-tuning (SFT) is insufficient. This is further demonstrated in our toy experiment that SFT cannot narrow the sampling distribution down sufficiently in Sec. A of our supplementary materials. We identified that **a comparative learning signal, driven by negative targets, is essential to enhance this ability** on both simple diffusion model sampling in toy experiments and complex subject-driven generation in the main paper, and we merely "borrow" the DPO formulation as an effective implementation of this novel learning strategy. To the best of our knowledge, we are the first to explicitly analyze and emphasize the role of negative targets for fine-tuning zero-shot subject-driven generation models.
>
> **2. The Non-trivial Extension due to Data Collection.**
>
> We further emphasize that **a "trivial extension" of Diffusion-DPO to our task is nontrivial in practice**. Diffusion-DPO typically learns from existing curated datasets with manually human-annotated pair-wise preference labels for general image quality. In contrast, for subject-driven generation, no such public dataset exists. The task **requires labels reflecting nuanced differences in subject fine-detail fidelity with respect to reference subjects, which are exceptionally difficult and prohibitively challenging to collect manually**. (It is worth noting that even collecting a Supervised Fine-Tuning (SFT) dataset is exceptionally challenging, as it requires gathering triplets of a subject image, a target text, and a corresponding target image. This difficulty is precisely why effective triplet dataset construction methods have emerged only very recently—lagging significantly behind the rapid advancements in text-to-image models—and why SFT itself has only just begun to demonstrate its full potential for this task.) Therefore, meaningful negative target construction becomes a critical and central component to implement our comparison-based fine-tuning. Our extensive ablation studies on dataset construction and their resulting performance differences validate this point.
>
> **3. Our Novel Contribution: CDNS.**
>
> This is precisely where **our key methodological contribution, CDNS, comes in**. CDNS is a novel and task-specific method for **automatically synthesizing target pairs with nuanced differences** in both subject fidelity and context from supervised fine-tuning dataset. It is what enables the successful application of the comparison-based learning formulation to the subject-driven generation task without the need for any manual labeling. Moreover, the resulting performance improvement varies drastically depending on the data construction strategy even when using the same comparative learning formulation, as supported by the table below (reorganized from Tab. 4 in the main paper) which presents the results of DPO formulation with various data construction strategies.
>
> |           | DINO  | CLIP-I | CLIP-T |
> |-----------|-------|--------|--------|
> | DPO-DINO  | 0.651 | 0.795  | 0.329  |
> | Self-Play | 0.685 | 0.799  | 0.326  |
> | CDNS      | 0.767 | 0.834  | 0.324  |
>
> This demonstrates that **simple DPO extension failed and the key element of our success is not the learning formulation itself, but rather the construction of effective negative targets via CDNS and their subsequent use in training.**
>
> We clarify this content more clearly in Sec. 2.2 of the main paper in the revised manuscript.

---

> > ### Author Response · Authors · 2025-11-21
> > **Rebuttal by Authors (2)**
> >
> > > ### **[W2] … the gain is marginal and not substantial enough to demonstrate a clear advantage in subject fidelity. Moreover, its CLIP-I and CLIP-T results remain below the state-of-the-art …**
> >
> > We appreciate the reviewer's detailed observation regarding the quantitative metrics. While the **numerical gains on the DINO score might appear marginal, this is only when compared against a few, highly competitive concurrent works such as Kontext (arXiv 2025.06). Moreover, we would like to clarify that Kontext achieves its performance using a non-public, 'in-house' dataset and its detailed training 'recipe' is entirely unknown**.
> > In contrast, our method demonstrates a clear and significant advantage over the majority of recently established baselines such as Diptych Prompting (CVPR 2025), EasyControl (ICCV 2025), and OminiControl (ICCV 2025). Most of all, we note that our SFO dramatically outperforms the base supervised fine-tuned model, OminiControl, by a large margin, demonstrating the effectiveness of incorporating negative targets.
> > With this context in mind, we would like to further highlight two critical points:
> >
> > * **Superiority in Human Evaluation and Qualitative Results**: We kindly ask the reviewer to consider our human evaluation results as in Tab. 2 of main paper, where our method achieves superior performance over all baselines. Notably, even against the strongest competitor, Kontext, our method achieved a 56% win rate in subject fidelity. Furthermore, our qualitative results in Fig. 4 of main paper clearly demonstrate that SFO yields notable improvements in fine-grained subject details and texture, which are areas where the baselines fail. We focus on this enhancement of fine-detail fidelity beyond automatic metrics as a crucial step toward the ultimate goal of subject-driven generation.
> > * **Achieving a Better Balance (Fidelity vs. Alignment)**: The reviewer’s concern about balancing subject fidelity with semantic/text alignment is precisely the core challenge. For CLIP-I (Image Alignment), we ask the reviewer to note that our method ranks second, surpassed only by UNO, with a negligible margin of only 0.001. Despite this nearly identical score, SFO outperforms UNO in other metrics and is remarkably preferred in human evaluation, achieving a 64.7% win rate against UNO in subject fidelity. For CLIP-T (Text Alignment), while SFO is not the SOTA, it remains highly competitive. More importantly, the baselines that score higher on CLIP-T (e.g., Diptych Prompting, OmniControl, EasyControl) suffer from significantly lower subject fidelity (as supported by both our DINO scores and qualitative examples).
> >
> > Therefore, we argue that SFO does not "struggle to balance" these aspects; rather, it **achieves a more appropriate and effective balance between subject fidelity and text alignment compared to prior work**, successfully improving fidelity without sacrificing text relevance.

---

> ### Author Response · Authors · 2025-11-21
> **Rebuttal by Authors (3)**
>
> > ### **[W3] … other diffusion backbones (SD-XL, Imagen) or broader subject-driven generation settings …**
>
> Thank you for raising this important point regarding cross-model generalization.
>
> Our initial motivation is to demonstrate our method's effectiveness on what we consider the most advanced, state-of-the-art architecture available, which leads us to choose the FLUX.
>
> However, we agree that demonstrating generalization is critical. To address your concern, we have conducted additional experiments. As the reviewer is aware, Imagen [1] is a closed-source model, making direct fine-tuning inaccessible.
> Therefore, we **apply our CDNS + SFO framework to a different backbone—IP-Adapter [2] on U-Net-based Stable Diffusion XL (SDXL)—** instead of the FLUX model (with its MM-DiT backbone) used in our main experiments.
> The results are as follows and our results demonstrate the generalizability of our comparison-based fine-tuning for subject-driven generation, which is appended to Sec. G of the Appendix.
>
> |                  | model | DINO  | CLIP-I | CLIP-T |
> |------------------|-------|-------|--------|--------|
> | IP-Adapter       | SD-XL | 0.613 | 0.810  | 0.292  |
> | IP-Adapter + SFO | SD-XL | 0.664 | 0.833  | 0.308  |
>
> With regard to other benchmarks, we first would like to note that DreamBench is the most widely adopted and representative benchmark composed of real-world subjects–serving as the primary and often sole benchmark for single subject-driven generation in most related literature.
>
> However, we agree with the reviewers' concern about broader setting and we **re-evaluate our method and our SFT baseline, OminiControl, on the CustomConcept101 dataset [3]**, which is more challenging and realistic than DreamBench, spanning diverse real-world subjects. For each concept, there are 20 prompts per concept that require background changes, insertion of new objects, style variations, or material/property changes of the subject. The results are as follows:
>
> |                    | DINO  | CLIP-I | CLIP-T |
> |--------------------|-------|--------|--------|
> | SFT (OminiControl) | 0.521 | 0.726  | 0.308  |
> | SFO                | 0.656 | 0.785  | 0.296  |
>
> On this diverse and challenging benchmark, our method also **significantly outperforms the supervised fine-tuned baseline regarding subject fidelity with comparable text alignment**, demonstrating the effectiveness of our CDNS-guided comparative learning strategy.
>
> We add this additional experimental results in Sec. 4.2 and Tab. 3 of the revised main paper, and Fig. H of the Appendix.
>
>
> > ### **[Q1] … Why does lower DINO similarity lead to better performance?**
>
> As demonstrated in DreamBooth [4] and subsequently adopted by many subject-driven generation papers, **DINO similarity is considered a reliable measurement for subject fidelity against a reference image**.
>
> The core principle of effective comparison-based learning in our method is to generate a learning signal from **the subject fidelity gap** between a positive and a negative target, teaching the model to favor the high-fidelity positive target. We measure this subject fidelity gap—which exists between the ground-truth positive target and our synthetically generated negative target—using DINO similarity. Therefore, **a lower DINO similarity can be interpreted as a more sufficient and clear fidelity gap, which is essential for generating a robust learning signal**.
>
> The negative targets generated by CDNS (which, as you noted, have lower DINO similarity values) create **a sufficiently large and clear fidelity gap**. Furthermore, it shows a wider spectrum of degradation levels as illustrated in Fig. 3 of the main paper. As a result, this generates sufficient fidelity gaps of varying difficulty, which provides the robust and effective learning signal necessary for the model to improve its fidelity.
>
> In contrast, the negative samples from the self-play method exhibit a highly-biased DINO similarity (i.e., they are often too similar to the positive target). This results in **an insufficient fidelity gap**, which fails to generate a strong or effective learning signal, as shown in row 4 of Tab. 4 in the main paper.
>
> ### **References**
>
> [1] Saharia et al., Photorealistic Text-to-Image Diffusion Models with Deep Language Understanding, NeurIPS 2022
>
> [2] Ye et al., IP-Adapter: Text Compatible Image Prompt Adapter for Text-to-Image Diffusion Models, arXiv
>
> [3] Kumari et al., Multi-Concept Customization of Text-to-Image Diffusion, CVPR 2023.
>
> [4] Ruiz et al., DreamBooth: Fine Tuning Text-to-Image Diffusion Models for Subject-Driven Generation, CVPR 2023.

---

### Official Review · Reviewer_TuuX · 2025-10-28

**Soundness:** 3
**Presentation:** 3
**Contribution:** 2
**Rating:** 4
**Confidence:** 4

**Summary:**

This paper focuses on enhancing subject fidelity in the field of subject-driven generation. It first identifies the inherent limitations of training under the supervised fine-tuning (SFT) framework, and accordingly proposes the Subject Fidelity Optimization (SFO) training framework. Concomitantly, it introduces the condition-degradation negative sampling (CDNS) strategy for generating negative samples. Both qualitative and quantitative experiments demonstrate the effectiveness of the proposed method.

**Strengths:**

- This paper is well-written and easy to follow.
- The method proposed in the paper is model-agnostic, meaning it can be applied for post-training on any base model to enhance subject fidelity.

**Weaknesses:**

- The CDNS presented in this paper demonstrates the ability to generate "negative" samples with significantly degraded quality. However, this raises concerns. As shown in Figure 3, some negative samples exhibit an excessive discrepancy in fidelity compared to the reference images and show little to no relevance to the text prompt. Such "positive-negative" samples may not provide effective learning signals for the model. This issue warrants further investigation.
- The proposed SFO training framework bears strong similarities to the diffusion-DPO method, and its novelty appears to be limited.
- Since the model is based on the OminiControl base model and fine-tuned through reinforcement learning, it would be beneficial to include comparisons with other reinforcement learning-based methods in the experimental section to provide a more comprehensive evaluation of its performance.

**Questions:**

- What are the differences between the proposed SFO method and the traditional DPO approach?
- Are there any other benchmarks in this field besides DreamBench? How does the method presented in this paper perform on these alternative benchmarks?

---

> ### Author Response · Authors · 2025-11-21
> **Rebuttal by Authors**
>
> Thank you very much for dedicating your time and effort to reviewing our manuscript.
>
> We would also like to express our sincere gratitude for your acknowledgment of the paper's overall writing quality and our proposed model-agnostic post-training approach.
> In the following, we will detail our responses to the concerns you raised and describe how we have revised the manuscript accordingly.
>
> > ### **[W1] … some negative samples exhibit an excessive discrepancy in fidelity compared to the reference images and show little to no relevance to the text prompt …**
>
> Regarding your concern about Fig. 3, we would like to clarify that the selected samples were chosen to demonstrate a stark contrast for illustrative purposes. However, as shown in the DINO similarity distribution (also in Fig. 3), our negative targets actually **span a wide spectrum of similarity relative to the positive targets, ranging from similar examples to dissimilar ones**. This distribution confirms that CDNS generates negative samples with various levels of fidelity degradation, not only the extreme cases.
> To empirically validate the impact of this diversity, we conduct an ablation study by categorizing the dataset into three subsets—High, Mid, and Low—based on the DINO similarity between positive and negative targets. We then train models on these subsets, and the experimental results are presented in the table below.
>
> |              | DINO  | CLIP-I | CLIP-T |
> |--------------|-------|--------|--------|
> | High+Mid+Low | 0.767 | 0.834  | 0.324  |
> | High         | 0.733 | 0.817  | 0.327  |
> | Mid          | 0.712 | 0.811  | 0.328  |
> | Low          | 0.698 | 0.801  | 0.327  |
> | SFT          | 0.652 | 0.795  | 0.329  |
>
> Aligning with your insight to some extent, the model trained solely on the 'Low' similarity subset—representing the cases with excessive discrepancy—shows the smallest improvement among SFO variants, although it still outperforms the SFT baseline. However, we observe progressively larger gains with the 'Mid' and 'High' subsets, and **the best performance is achieved when utilizing the full spectrum of diversity (High+Mid+Low)**. These results demonstrate that exposing the model to **a comprehensive range of difficulties, facilitated by our proposed CDNS, contributes to a significantly more robust and effective training process**.
> We add this content and additional experimental results in Sec. 3.3 of the main paper and Sec. D.6 and Tab. F of the Appendix.
>
> Furthermore, our text prompt degradation process also generates a diverse range of contexts. We discover that **this comprehensive signal effectively improves the performance while maintaining its alignment with the text prompt**, as demonstrated by the results in the table below, which reorganizes rows 1 and 4 from Tab. 5 in the main paper.
>
> |                     | DINO  | CLIP-I | CLIP-T |
> |---------------------|-------|--------|--------|
> | SFO w/ text degrad  | 0.767 | 0.834  | 0.324  |
> | SFO w/o text degrad | 0.733 | 0.826  | 0.325  |

---

> > ### Author Response · Authors · 2025-11-21
> > **Rebuttal by Authors (2)**
> >
> > > ### **[W2] … strong similarities to the diffusion-DPO method …**
> > > ### **[Q1] … differences between the proposed SFO method and the traditional DPO approach …**
> >
> > While our method may appear to share similarities with the DPO formulation, we clarify our contributions go beyond a direct application, centered on a different motivation and a novel, non-trivial data construction method applying for subject-driven generation. We further emphasize that the formulation alone is insufficient, as evidenced by our dataset construction ablation study; our negative targets synthesized from CDNS is a prerequisite for unlocking the potential of this formulation.
> >
> > **1. Difference in Motivation and Objective.**
> >
> > While existing Diffusion-DPO work focuses on improving general image quality, our primary objective is fundamentally different: we **aim to enhance the model's "mode-narrowing" ability**.
> > We argue that for tasks like subject-driven generation—which require sampling from a very specific mode (i.e., matching the subject)—simple supervised fine-tuning (SFT) is insufficient. This is further demonstrated in our toy experiment that SFT cannot narrow the sampling distribution down sufficiently in Sec. A of our supplementary materials. We identified that **a comparative learning signal, driven by negative targets, is essential to enhance this ability** on both simple diffusion model sampling in toy experiments and complex subject-driven generation in the main paper, and we merely "borrow" the DPO formulation as an effective implementation of this novel learning strategy. To the best of our knowledge, we are the first to explicitly analyze and emphasize the role of negative targets for fine-tuning zero-shot subject-driven generation models.
> >
> > **2. The Non-trivial Extension due to Data Collection.**
> >
> > We further emphasize that **a "trivial extension" of Diffusion-DPO to our task is nontrivial in practice**. Diffusion-DPO typically learns from existing curated datasets with manually human-annotated pair-wise preference labels for general image quality. In contrast, for subject-driven generation, no such public dataset exists. The task **requires labels reflecting nuanced differences in subject fine-detail fidelity with respect to reference subjects, which are exceptionally difficult and prohibitively challenging to collect manually**. (It is worth noting that even collecting a Supervised Fine-Tuning (SFT) dataset is exceptionally challenging, as it requires gathering triplets of a subject image, a target text, and a corresponding target image. This difficulty is precisely why effective triplet dataset construction methods have emerged only very recently—lagging significantly behind the rapid advancements in text-to-image models—and why SFT itself has only just begun to demonstrate its full potential for this task.) Therefore, meaningful negative target construction becomes a critical and central component to implement our comparison-based fine-tuning. Our extensive ablation studies on dataset construction and their resulting performance differences validate this point.
> >
> > **3. Our Novel Contribution: CDNS.**
> >
> > This is precisely where **our key methodological contribution, CDNS, comes in**. CDNS is a novel and task-specific method for **automatically synthesizing target pairs with nuanced differences** in both subject fidelity and context from supervised fine-tuning dataset. It is what enables the successful application of the comparison-based learning formulation to the subject-driven generation task without the need for any manual labeling. Moreover, the resulting performance improvement varies drastically depending on the data construction strategy even when using the same comparative learning formulation, as supported by the table below (reorganized from Tab. 4 in the main paper) which presents the results of DPO formulation with various data construction strategies.
> >
> > |           | DINO  | CLIP-I | CLIP-T |
> > |-----------|-------|--------|--------|
> > | DPO-DINO  | 0.651 | 0.795  | 0.329  |
> > | Self-Play | 0.685 | 0.799  | 0.326  |
> > | CDNS      | 0.767 | 0.834  | 0.324  |
> >
> > This demonstrates that **simple DPO extension failed and the key element of our success is not the learning formulation itself, but rather the construction of effective negative targets via CDNS and their subsequent use in training.**
> >
> > We clarify this content more clearly in Sec. 2.2 of the main paper in the revised manuscript.

---

> ### Author Response · Authors · 2025-11-21
> **Rebuttal by Authors (3)**
>
> > ### **[W3] … comparisons with other reinforcement learning-based methods …**
>
> We appreciate the reviewer's suggestion regarding reinforcement learning-based formulation to realize comparative learning. While our SFO adopts the DPO formulation to implement comparative learning signals, we understand the value of comparing it with online RL methods to strictly validate this design choice. To address this, we **implement a robust Online RL baseline tailored for our task and compare it to SFO**.
> To construct a fair RL baseline, we first train a reward model from scratch using the quadruplet dataset including CDNS negatives. We modify the architecture of ImageReward [1], a standard text-to-image human preference reward model, to accept an additional reference image condition, enabling it to evaluate subject fidelity alongside text alignment. Leveraging this trained reward model, we apply an online RL algorithm adapted for Flow Matching models. Specifically, we utilize the DDPO (Denoising Diffusion Policy Optimization) [2] methodology to perform RL-based preference optimization.
>
> |                    | DINO  | CLIP-I | CLIP-T |
> |--------------------|-------|--------|--------|
> | SFT (OminiControl) | 0.652 | 0.795  | 0.329  |
> | SFO                | 0.767 | 0.834  | 0.324  |
> | Reward + DDPO      | 0.641 | 0.790  | 0.330  |
>
> The comparison results are presented in the table above. Unlike our SFO (which treats the generative model as an implicit reward model and learns directly from data), **the explicit Reward Model-based Online RL approach yielded results inferior to the baseline**.
>
> We analyze the reasons as follows:
>
> * **Dependency on Reward Model Quality**: Subject-driven generation requires evaluating fine-grained details (subject fidelity) rather than coarse image quality. Training a reward model to accurately capture these subtle nuances proves to be exceptionally difficult compared to general quality assessment. Consequently, the limitation of the explicit reward model becomes a bottleneck.
>
> * **Ineffective Exploration & Sparse Signals**: In the online RL setting where the model acts as a policy, we observe that the model mostly generates samples that receive high rewards from the imperfect reward model (similar to the self-play limitation). This results in sparse negative feedback, failing to provide the effective learning signal for mode-narrowing.
>
> Despite utilizing effective negative targets from CDNS, the instability of Online RL and the difficulty of training a precise reward model limit its applicability to this task. Furthermore, Online RL requires iterative sampling at every training step, which is computationally intensive and inefficient (e.g., Flow-GRPO [3] reportedly requires 100-200 GPU hours on H800). These results highlight that **synergy between our SFO formulation and the negative targets tailored for subject-driven generation via CDNS is currently the most effective and efficient comparative learning strategy for subject-driven generation**. We believe advancing reward models and Online RL algorithms for such fine-grained mode-narrowing tasks in TTI models remains a challenging direction for future work.
> We append these additional ablation results to Sec. 4.3 of the revised manuscript.
>
> > ### **[Q2] …  other benchmarks in this field besides DreamBench …**
>
> While we note that DreamBench is the representative benchmark composed of real-world subjects–serving as the primary and often sole benchmark for single subject-driven generation in most related literature— we also **recognize CustomConcept101 [4] as a valuable alternative benchmark to address your concern**.
> CustomConcept101 is more challenging and realistic than DreamBench, spanning diverse real-world subjects. For each concept, there are 20 prompts per concept that require background changes, insertion of new objects, style variations, or material/property changes of the subject.
>
> We compare our SFO and SFT baseline, OminiControl, on this alternative benchmark and the results are as follows:
>
> |                    | DINO  | CLIP-I | CLIP-T |
> |--------------------|-------|--------|--------|
> | SFT (OminiControl) | 0.521 | 0.726  | 0.308  |
> | SFO                | 0.656 | 0.785  | 0.296  |
>
> On this diverse and challenging benchmark, our method also **significantly outperforms the SFT baseline regarding subject fidelity with comparable text alignment, demonstrating the effectiveness of our CDNS-guided comparative learning strategy**.
> We add this additional experimental results in Sec. 4.2 and Tab. 3 of the revised main paper, and Fig. H of the Appendix.
>
> ### **References**
>
> [1] Xu et al., ImageReward: Learning and Evaluating Human Preferences for Text-to-Image Generation, NeurIPS 2023
>
> [2] Black et al., Training Diffusion Models with Reinforcement Learning, ICLR 2024
>
> [3] Liu et al., Flow-GRPO: Training Flow Matching Models via Online RL, NeurIPS 2025
>
> [4] Kumari et al., Multi-Concept Customization of Text-to-Image Diffusion, CVPR 2023.

---

> > ### Comment · Reviewer_TuuX · 2025-11-22
> >
> > Thank you for providing a comprehensive rebuttal, which includes new experimental results and comparisons. The authors' responses have successfully addressed **some** of my initial concerns regarding the efficacy and robustness of the proposed method.
> >
> > Specifically, the inclusion of the comparison against DDPO and the positive results on the CustomConcept101 benchmark further substantiate the method's effectiveness and practical advantages over alternative reinforcement learning approaches.
> >
> > However, I still have reservations concerning the fundamental claims.
> >
> > First
> >
> > The core motivation of the proposed CDNS technique, as stated in line 269 of the manuscript, is:
> >
> > > “In particular, naive Self-Play generates negatives by conditioning on the same inputs as the positives, often yielding nearly identical contexts and insufficient fidelity discrepancies (see row 3 of Fig. 3(a)).
> > >
> > > CDNS deliberately applies degradation in both image and text conditioning modalities during negative data synthesis to induce further discrepancies.”
> >
> > However, the new ablation study presented in the rebuttal suggests a contradiction:
> >
> > - The model trained solely on the **'High' similarity subset** exhibits the largest improvement.
> > - The argument that *"the best performance is achieved when utilizing the full spectrum of diversity (High+Mid+Low)*" could be attributed to the **increased data size** provided by this combination.
> >
> > This leads to a key logical ambiguity: If the 'High' similarity subset yields the best result, the primary design motivation of CDNS (deliberately induce large discrepancies) appears to be **unjustified**.
> >
> > Second
> >
> > My concern regarding the strong similarities between SFO and the existing Diffusion-DPO method remains. While the authors correctly highlight a difference in objective and the challenge of data collection, the underlying framework appears to be same. Therefore, the novelty of the SFO remains limited.
> >
> > However, I want to emphasize that the direction of the authors' data collection effort is highly insightful. The lack of specific data for subject-driven generation makes the simple application of DPO non-trivial. From this perspective, the design of CDNS itself is both novel and reasonable as a critical enabling technology to synthesize the necessary paired data for this challenging task.
> >
> > Based on the above, I choose to keep my score but lower my confidence.

---

> ### Author Response · Authors · 2025-11-24
> **Rebuttal by Authors (4)**
>
> We sincerely thank the reviewer for the constructive feedback. We greatly appreciate the opportunity to further discuss these fundamental aspects of our work.
>
> **1. Regarding the "Key Logical Ambiguity" (Data Size & Motivation)**
>
> We appreciate you highlighting the potential factor of dataset size in our analysis. To address this rigorous point, we conducted a new controlled experiment where we **equalized the total number of training samples across all settings**. Specifically, we expanded the CDNS generation and categorized the synthesized negative targets into 'High', 'Mid', and 'Low' sets based on their DINO similarity ranges to match the total data count of the 'High+Mid+Low' setting.
> The results are as follows:
>
> |                       | DINO  | CLIP-I | CLIP-T |
> |-----------------------|-------|--------|--------|
> | Very High (Self-Play) | 0.685 | 0.799  | 0.326  |
> | High + Mid + Low      | 0.767 | 0.834  | 0.324  |
> | High                  | 0.723 | 0.811  | 0.329  |
> | Mid                   | 0.741 | 0.815  | 0.327  |
> | Low                   | 0.723 | 0.815  | 0.328  |
> | SFT                   | 0.652 | 0.795  | 0.329  |
>
> As shown in the table, even when the data size is controlled, **the Full Spectrum setting (High+Mid+Low) achieves the best performance compared to any single set (High, Mid, or Low)**. This confirms that exposing the model to diverse levels of fidelity degradation is indeed more effective than training on a specific narrow range.
>
> Moreover, we would like to clarify the definition of our 'High' set to resolve the perceived contradiction. Unlike the Self-Play method, which generates negatives with "very high" similarity (averaging DINO Similarity 0.82), **our CDNS 'High' set still undergoes degradation (averaging DINO Similarity 0.77)**. Therefore, compared to Self-Play samples, our 'High' samples possess a small but meaningful fidelity gap, still serving as effective negatives that refine the model's discrimination ability. Consequently, the primary motivation of CDNS is not merely to maximize discrepancy (which would correspond only to the 'Low' set), but to generate a diverse spectrum of sufficient fidelity gaps compared to Self-Play. While we specify as “High” similarity, **CDNS covers a range extending from sufficiently moderate differences (“High”) to highly distinct deviations (“Low”)**, allowing the model to learn from comprehensive difficulty levels, leading to the most robust performance.
>
> **2. Regarding the Novelty of SFO Framework**
>
> First, we are deeply grateful for your recognition of our CDNS design as "highly insightful," "novel," and a "critical enabling technology."
>
> Regarding your concern about the formulation, we respectfully wish to highlight the current landscape of research in this domain. Diffusion-DPO was indeed a ground-breaking study that established the mathematical foundation for preference optimization in diffusion models. **Following this, recent top-tier research focuses on effectively adapting and tailoring this formulation to solve specific downstream tasks where standard application fails. Prominent examples** recognized in major conferences include: DPO for per-subject fine-tuning-based subject-driven generation [1] in NeurIPS 2024 (which we compare in our Appendix), DPO for human image generation [2] in CVPR 2025, DPO for text-to-design image generation [3] in CVPR 2025, and DPO for 3D structure prediction [4] in NeurIPS 2024.
>
> Our work follows this established research trajectory. We identified that applying DPO to zero-shot subject-driven generation is non-trivial due to the lack of paired data. **Our SFO (with CDNS) is a tailored solution designed to unlock this capability, a necessity underscored by our finding that a naive DPO application fails**. In this context, we believe our contribution lies in successfully bridging the gap between the DPO formulation and the zero-shot subject-driven generation task through the novel design of CDNS.
>
> **References**
>
> [1] Miao et al., Subject-driven Text-to-Image Generation via Preference-based Reinforcement Learning, NeurIPS 2024
>
> [2] Na et al., Boost Your Human Image Generation Model via Direct Preference Optimization, CVPR 2025
>
> [3] Wang et al., DesignDiffusion: High-Quality Text-to-Design Image Generation with Diffusion Models, CVPR 2025
>
> [4] Jiao et al., 3D Structure Prediction of Atomic Systems with Flow-Based Direct Preference Optimization, NeurIPS 2024

---

### Official Review · Reviewer_68io · 2025-11-01

**Soundness:** 3
**Presentation:** 4
**Contribution:** 3
**Rating:** 8
**Confidence:** 4

**Summary:**

This paper focuses on the subject fidelity problem in zero-shot subject-driven generation and makes the following contributions:
(1) Subject Fidelity Optimization (SFO) guides the model to favor positive samples over negatives through pairwise comparison.
(2) Condition-Degradation Negative Sampling (CDNS) synthesizes negative targets tailored for subject-driven generation in a controlled setting.

Through these contributions, the proposed method demonstrates strong results in improving subject fidelity.

**Strengths:**

Well-written and easy to read and follow.

A simple, well-designed, and effective method pipeline — particularly, the Condition-Degradation Negative Sampling pipeline is a strong design choice.

Strong experimental results.

**Weaknesses:**

Testing on more benchmarks could further strengthen the evidence for the method’s effectiveness.
No other clear weaknesses.

**Questions:**

Suggestions:
The color scheme of some figures could be further optimized for better readability.

---

> ### Author Response · Authors · 2025-11-21
> **Rebuttal by Authors**
>
> We sincerely appreciate your positive feedback on the effort dedicated to our paper's writing, the overall design choices and pipeline structure, and the experimental results.
>
> > ###  **[W1] Testing on more benchmarks ...**
>
> Regarding the concern about more benchmarks, we first would like to note that DreamBench is the most widely adopted and representative benchmark composed of real-world subjects–serving as the primary and often sole benchmark for single subject-driven generation in most related literature.
>
> However, we also **recognize CustomConcept101 [1] as a valuable alternative benchmark**. CustomConcept101 is more challenging and realistic than DreamBench, spanning diverse real-world subjects. For each concept, there are 20 prompts per concept that require background changes, insertion of new objects, style variations, or material/property changes of the subject.
> We demonstrate the effectiveness of our method on this alternative benchmark by comparing our SFT baseline, OminiControl. The results are as follows:
>
> |                    | DINO  | CLIP-I | CLIP-T |
> |--------------------|-------|--------|--------|
> | SFT (OminiControl) | 0.521 | 0.726  | 0.308  |
> | SFO                | 0.656 | 0.785  | 0.296  |
>
> On this diverse and challenging benchmark, our method also **significantly outperforms the SFT baseline regarding subject fidelity with comparable text alignment**, demonstrating the effectiveness of our CDNS-guided comparative learning strategy.
>
> We add this additional experimental results in Sec. 4.2 and Tab. 3 of the revised main paper, and Fig. H of the Appendix.
>
> ### **References**
>
> [1] Kumari et al., Multi-Concept Customization of Text-to-Image Diffusion, CVPR 2023.

---

### Author Response · Authors · 2025-11-30
**Summary for the Newly Assigned Area Chair**

We sincerely appreciate the time and effort you have dedicated to reviewing our work. To assist you as the newly assigned Area Chair in navigating our submission more efficiently, we would like to provide a concise summary of our key contributions and the rebuttal process.

First, our method proposes the introduction of negative targets and a comparative learning framework to enhance zero-shot subject-driven generation in diffusion models. Crucially, we enabled this comparative learning through the proposal of Condition-Degradation Negative Sampling (CDNS), an effective method for synthesizing negative targets.

Reviewers have recognized our work as **well-written and structured** (Reviewers 68io, TuuX, ajgK), presenting **a model-agnostic, simple, and well-designed framework** (Reviewers 68io, TuuX, 3v7S) backed by **extensive experimental results** (Reviewers 68io, 3v7S).

Regarding the concerns raised during the discussion period, we have provided the following responses:

* **Novelty**: While some concerns were raised regarding the similarity of our formulation to DPO, we clarified that while Diffusion-DPO [1] aims to improve general image quality, our primary objective is to **strengthen the "mode-narrowing ability" essential for subject-driven generation**. Furthermore, a direct extension of Diffusion-DPO is hindered by the lack of paired datasets—a significant challenge where simple implementations fail. Our key contribution lies in effectively and efficiently realizing this through CDNS. Therefore, we highlight that our primary contribution is **the effective dataset construction strategy and the successful realization of comparative learning, rather than the formulation itself**. This aligns with recent trends in top-tier conferences (e.g., CVPR, NeurIPS), where the core contribution often lies in the effective adaptation of the DPO framework to specific tasks rather than the formulation itself [2,3,4,5].

* **Single Benchmark & Backbone Model**: Beyond the commonly used DreamBench, we demonstrated that our method **outperforms the supervised fine-tuned baseline on CustomConcept101** [6], a point acknowledged by Reviewer TuuX. We also verified our method's applicability beyond the state-of-the-art Flux model by demonstrating **its effectiveness on the IP-Adapter** [7] based on a different backbone (SD-XL).

* **Additional Ablation**: We further validated the effect of CDNS's diversity level through DINO similarity distribution analysis and strengthened the rationale for our comparative learning formulation design choice through comparisons with other reinforcement learning approaches [8].

We hope this summary provides a comprehensive overview of our rebuttal process. Beyond this summary, we kindly ask you to review our specific responses provided during the rebuttal process and revised manuscript for comprehensive details.

### **References**

[1] Wallace et al., Diffusion Model Alignment Using Direct Preference Optimization, CVPR 2024

[2] Miao et al., Subject-driven Text-to-Image Generation via Preference-based Reinforcement Learning, NeurIPS 2024

[3] Jiao et al., 3D Structure Prediction of Atomic Systems with Flow-Based Direct Preference Optimization, NeurIPS 2024

[4] Na et al., Boost Your Human Image Generation Model via Direct Preference Optimization, CVPR 2025

[5] Wang et al., DesignDiffusion: High-Quality Text-to-Design Image Generation with Diffusion Models, CVPR 2025

[6] Kumari et al., Multi-Concept Customization of Text-to-Image Diffusion, CVPR 2023

[7] Ye et al., IP-Adapter: Text Compatible Image Prompt Adapter for Text-to-Image Diffusion Models, arXiv

[8] Black et al., Training Diffusion Models with Reinforcement Learning, ICLR 2024

---

### Meta-Review · Area_Chair_8SpM · 2026-01-07

**Summary:**

This paper proposes a pipeline for zero-shot diffusion customization by combining SFT with pairwise learning. The pairwise learning part, termed as subject fidelity optimization, bears similarity with the standard diffusion-DPO framework.

Three reviewers have raised major concerns about the novelty w.r.t. diffusion-DPO. The authors have clarified that this paper is the first to apply DPO to the specific application of zero-shot subject-driven generation. The authors also claimed the main novelty is indeed the part for synthesizing negative samples, which is non-trivial.

The proposed method is technically sound. However, the novelty is limited as pointed out by several reviewers. The AC does not recommend accepting this paper. It can be more suited for venues that focus less on novelty but more on correctness.

**Reviewer Concerns:**

Concerns that were addressed by the rebuttal:
- Results on more benchmarks (Reviewer 68io, TuuX). New results on CustomConcept101 benchmark is provided in the rebuttal.
- The proposed framework with negative and positive pairs is similar to DPO based reinforcement learning method. (Reviewer TuuX). The authors have clarified the difference between DPO and the advantage over DPO. The concern is PARTIALLY addressed, as the reviewer acknowledged the contribution of negative data synthesis, but still thinks the formulation is extremely similar to DPO.
- Comparison to other reinforcement learning methods (Reviewer TuuX). The authors have implemented other RL methods which are shown to be inferior to the proposed framework in the rebuttal.
- Improvement over baselines is marginal (Reviewer 3v7S)
- Performance on other backbones (Reviewer 3v7S)

Outstanding concerns:
- Similarity between the objective here and diffusion-DPO. The novelty is limited. (Reviewer TuuX, 3v7S, ajgK)

**Reviewer Scores:**

Reviewer 68io is likely to keep their initial score of 8. However, the review is extremely short.

Reviewer TuuX has indicated keeping their score of 4.

Reviewer 3v7S is likely to keep their initial score of 4.

Reviewer ajgK is likely to keep their initial score of 4.

---

### Decision · Program_Chairs · 2026-01-26

Reject